# LATENT DIFFUSION-BASED 3D MOLECULAR RECOVERY FROM VIBRATIONAL SPECTRA

## ABSTRACT

Infrared (IR) spectroscopy, a type of vibrational spectroscopy, is widely used for molecular structure determination and provides critical structural information for chemists. However, existing approaches for recovering molecular structures from IR spectra typically rely on one-dimensional SMILES strings or two-dimensional molecular graphs, which fail to capture the intricate relationship between spectral features and three-dimensional molecular geometry. Recent advances in diffusion models have greatly enhanced the ability to generate molecular structures in 3D space. Yet, no existing model has explored the distribution of 3D molecular geometries corresponding to a single IR spectrum. In this work, we introduce `IR-GeoDiff`, a latent diffusion model that integrates IR spectral information into both node and edge representations of molecular structures. We evaluate IR-GeoDiff from both spectral and structural perspectives, demonstrating its ability to recover the molecular distribution corresponding to a given IR spectrum. Furthermore, an attention-based analysis reveals that the model's interpretation of IR spectra aligns with quantum mechanical principles of molecular vibrations.

## 1 INTRODUCTION

Chemists have developed various molecular spectroscopic methods to elucidate molecular structures, among which infrared (IR) spectroscopy is widely applied in different scenarios (Sun, 2009; Tinetti et al., 2013; Muro et al., 2015; Renner et al., 2019) and is capable of identifying various types of compounds (Becucci & Melandri, 2016; Hadjiivanov et al., 2020; Adhikary et al., 2017). IR spectra originate from molecular vibrations, and peaks appearing in specific spectral regions correspond to particular vibrational modes. Chemists typically interpret IR spectra by identifying characteristic bonds or functional groups based on empirical rules (Coates et al., 2000) or through visual comparisons with reference spectra from standard databases (Zhou et al., 2012) or computational simulations (Henschel et al., 2020; Esch et al., 2021). However, IR spectra often exhibit complex peak patterns, especially in the fingerprint region (400-1,500 cm$^{-1}$), posing significant interpretation challenges. Furthermore, limited database coverage and variability of experimental conditions constrain the effectiveness of visual comparison methods.

Researchers have applied artificial neural networks to elucidate IR spectra of small organic molecules since the 1990s (Gasteiger & Zupan, 1993; Burns & Whitesides, 1993; Hemmer et al., 1999), the primary task of which was classifying predefined functional groups (Gasteiger & Zupan, 1993; Burns & Whitesides, 1993; Judge et al., 2008; Fine et al., 2020; Wang et al., 2020; Enders et al., 2021; Jung et al., 2023; Wang et al., 2023; Lu et al., 2024; Rieger et al., 2023), the specific groups of atoms bonded in certain arrangements that together exhibit their own characteristic properties. Recently, this task has been advanced with Transformer models employed to generate complete molecular structures directly from IR spectra along with chemical formulas, achieving notable performance (Alberts et al., 2024a; Wu et al., 2025; Alberts et al., 2025). In these studies, molecular structures are represented as 1D representations in terms of SMILES (Simplified Molecular Input Line Entry System) (Weininger, 1988) strings. Although SMILES strings well encode structural information through developed grammar rules, they do not provide explicit interatomic relationships. Moreover, a single molecular structure can correspond to multiple valid SMILES strings, which introduces representational ambiguity and requires models to implicitly learn the underlying grammar. No matter whether molecules are represented as 1D SMILES strings or 2D molecular graphs, essential 3D structural information is inevitably lost or severely reduced, despite IR spectra originating from molecular vibrations, which are inherently 3D phenomena reflecting atomic spatial arrangements.

This representation gap makes it difficult to understand how the model interprets spectral features accurately. However, using 3D geometries to represent molecular structure requires the model to have a deeper understanding of molecular structure to complement 3D spatial information and generate valid molecules.

Recently, diffusion models (DMs) have emerged as powerful tools for exploring 3D molecular geometries, facilitating tasks such as *de novo* molecular generation (Alakhdar et al., 2024), conformer generation (Jing et al., 2022) and molecular dynamics (Wu & Li, 2023; Han et al., 2024). Despite these advances, existing works, to a large extent, have overlooked molecular spectroscopy, an essential approach for elucidating 3D molecular structures in the real world. Directly generating molecular structures from infrared spectra holds significant practical value. If highly automated structural interpretation becomes feasible, it could greatly accelerate molecular screening and validation workflows in real-world applications such as materials design (Lansford & Vlachos, 2020). Different spectroscopies contain different molecular information, and each individual spectroscopy can be seen as a dimensionally reduced representation of the full 3D structures. In this paper, we propose a new **IR** spectra-guided **Geo**metric latent **Diff**usion model (`IR-GeoDiff`) to recover the distribution of molecular geometries represented by IR spectroscopy. As molecular vibrations primarily affect the interatomic distance, we not only incorporate IR spectral information into atomic features, but also the edges between any two atoms within the molecule. By visualising the cross-attention map between the features of those edges and spectral patches, we found that the way in which our model analyses spectra and structure aligns with the quantum theories of IR spectroscopy. In summary, our contributions are as follows:

1. We introduce a new task of recovering the distribution of 3D molecular geometries encoded within infrared spectroscopy. This formulation bridges molecular structure generation and spectroscopic analysis, and highlights the intrinsic link between vibrational spectral patterns and 3D molecular geometries.

2. To the best of our knowledge, `IR-GeoDiff` is the first model to directly recover 3D molecular geometries from 1D infrared spectra, introducing a new paradigm for the automatic interpretation of IR spectra by leveraging 3D diffusion models.

3. We propose a comprehensive set of evaluation metrics that assess whether the recovered molecular structures are consistent with the distribution encoded in the input infrared spectra, from both structural and spectral perspectives.

4. We further investigate how the proposed model works, and find that its behaviour is consistent with how human chemists interpret IR spectra and aligns with quantum chemical calculations of molecular vibrations, offering insights into model interpretability.

## 2 RELATED WORK

**IR spectra to molecular structure.** Computer-aided interpretation of IR spectra can be categorised into expert systems, library-search methods, and direct structure prediction approaches (Jung et al., 2023; Xue et al., 2023). Traditionally, researchers have applied various machine learning models, including multi-layer perceptrons (MLPs) (Fine et al., 2020), support-vector machines (SVMs) (Wang et al., 2020), and convolutional neural networks (CNNs) (Enders et al., 2021; Jung et al., 2023; Wang et al., 2023; Lu et al., 2024; Rieger et al., 2023), to classify specific substructures (i.e., functional groups) within molecules based on IR spectra. Regarding the direct prediction of entire molecular structures from IR spectra, Hemmer et al. (1999) proposed a method to derive 3D molecular structures from IR spectra by representing molecules as radial distribution function (RDF) codes. However, their method depended on database retrieval to reconstruct molecular geometries from RDF codes. After this early attempt, the task of directly predicting molecular structures received limited attention until recently. Ellis et al. (2023) revisited this problem, developing a deep Q-learning-based model for 2D molecular graph generation. More recently, Transformer-based methods have introduced substantial advances, directly generating molecular structures as 1D SMILES strings from IR spectra and demonstrating remarkable performance on both simulated and experimental datasets (Alberts et al., 2024a; Wu et al., 2025; Alberts et al., 2025). However, these methods generate 1D SMILES strings or 2D graphs rather than full 3D geometries, limiting their ability to capture the 3D structural information that underlies vibrational spectra. In contrast, IR-GeoDiff directly explore the relationship between 3D molecular geometries and IR spectra. The spectral information is injected into both atomic-level node features and bond-level edge features. In addition, inspired by prior work on

functional group classification from IR spectra, we further incorporate functional group features extracted from the spectra into the edge representations. Taken together, these conditioning signals more faithfully reflect the physical origin of IR absorption in the vibrations of specific bond and local patterns.

**Diffusion models for 3D molecular generation.** Diffusion models (Sohl-Dickstein et al., 2015b; Ho et al., 2020) have attracted significant attention due to their remarkable performance in generative tasks across diverse fields, including computer vision (Ho et al., 2020; Rombach et al., 2022), natural language processing (Austin et al., 2021; Hoogeboom et al., 2021), and molecular generation in both 2D (Vignac et al., 2023; Jo et al., 2022) and 3D (Hoogeboom et al., 2022; Xu et al., 2022; Wu et al., 2022; Xu et al., 2023; Huang et al., 2023; O Pinheiro et al., 2023; Qiang et al., 2023) space. Hoogeboom et al. (2022) introduced E(3)-equivariant diffusion models (EDMs) for molecular generation tasks in 3D space. Building upon this work, Xu et al. (2023) proposed geometric latent diffusion models (GEOLDM), utilising latent space representations to model complex molecular geometries. In the molecular domain, desired conditions typically are chemical properties, such as polarizability and orbital energies, thus simply concatenating to node features can realise conditional generation(Hoogeboom et al., 2022; Xu et al., 2023). More complex conditions for drug design, such as protein pockets (Schneuing et al., 2024; Guan et al., 2023; Peng et al., 2022) and reference molecular structures (Chen et al., 2023; Adams et al., 2025), often require either customised neural network architectures or specialised sampling strategies (Peng et al., 2022; Ayadi et al., 2024) to effectively incorporate conditional information into diffusion models. Recently, Cheng et al. (2024) proposed KREED, which determines 3D molecular structures from rotational spectra by leveraging the molecular formula, Kraitchman's substitution coordinates, and principal moments of inertia. Bohde et al. (2025) introduced DiffMS, which reconstructs molecules represented as 2D graphs from mass spectra, given the molecular formula. Despite this progress, there is currently no diffusion model that is conditioned on full IR spectra and learns a distribution over three-dimensional molecular geometries. Likewise, no evaluation protocol has been established for this new spectrum-to-geometry recovery task, which is fundamentally different from de novo molecular generation or conformer generation. The objective is not to encourage diversity in the generated molecules, but to recover geometries that are consistent with a given spectrum and to reduce the candidate space as much as possible. To address these challenges, we design IR-GeoDiff as an SE(3)-equivariant latent diffusion model conditioned on spectral features and functional group features. A Transformer-based spectral encoder extracts global spectral representations along with chemically meaningful functional group features, which serve as conditioning signals for the denoising network. These conditioning signals are integrated into the latent diffusion process via multi-head cross-attention. This design is tailored to the physics of IR spectroscopy, and it provides a more interpretable way to fuse spectral and geometric features. In addition, we propose a comprehensive set of evaluation metrics that assess whether the recovered geometries are consistent with the input IR spectra from both structural and spectral perspectives.

## 3 PRELIMINARIES

### 3.1 PROBLEM DEFINITION

The molecular geometry is represented as $G = \langle \mathbf{x}, \mathbf{h} \rangle$, where $\mathbf{x} = (\mathbf{x}_1, \ldots, \mathbf{x}_N) \in \mathbb{R}^{N \times 3}$ denotes the 3D Cartesian coordinates of atoms, and $\mathbf{h} = (h_1, \ldots, h_N) \in \mathbb{R}^{N \times 1}$ denotes their atomic types. Denote the input infrared spectrum as $S \in \mathbb{R}^{1 \times l}$, where $l$ is the number of sampled spectral points. Our goal is to learn a probabilistic model $\theta$ that captures the conditional distribution of molecular geometries given an IR spectrum, i.e. $p_\theta(G|S)$. Unlike conventional conditional molecular generation tasks, which aim to produce diverse structures under certain constraints like polarizability and protein pockets, we focus on recovering the set of 3D molecular geometries consistent with a given IR spectrum, where the goal is not to encourage diversity, but to reduce the candidate space as much as possible. In principle, the IR spectrum may correspond to a unique molecular structure, and the conditional distribution could collapse to a single geometry.

In practical experimental workflows, the interpretation of IR spectra is typically performed after the molecular formula, including the number and types of atoms, has been determined (Kind & Fiehn, 2007; Field et al., 2013); for example, through elemental analysis or complementary spectroscopic techniques. This is because IR spectra primarily encode vibrational information related to chemical bonds and functional groups rather than atomic identity. Therefore, in our setting, we assume that the

atom types $\mathbf{h}$ and the atom count $N$ of a molecule are known, and focus on modelling the conditional distribution over atomic coordinates $\mathbf{x}$. In other words, our problem can be formulated as $p_\theta(\mathbf{x}|S, \mathbf{h})$.

## 3.2 DIFFUSION MODELS

Diffusion models (Sohl-Dickstein et al., 2015a; Ho et al., 2020) are probabilistic models modelling two processes, a diffusion process and a denoising process, which can be described as the forward and reverse processes of a Markov chain with a fixed length $T$. In the forward process, the data sample x is mapped to a series of intermediate variables $x_1 \ldots x_T$ of the same dimensionality by gradually adding noise $\epsilon$ sampled from a standard normal distribution,

$$q(x_t|x_{t-1}) = \mathcal{N}(x_t; \sqrt{1-\beta_t}x_{t-1}, \beta_t I), \tag{1}$$

where $\beta_t$ is a hyperparameter controlling the extent of noise blending. With sufficient steps $T$, the distribution of variable $x_T$ converges to a standard normal distribution, $q(x_T) \approx \mathcal{N}(0, I)$, which is also the initial state of the denoising process. The true reverse distribution $q(x_{t-1}|x_t)$ is approximated as normal distributions,

$$p_\theta(x_{t-1}|x_t) = \mathcal{N}(x_{t-1}; \mu_\theta(x_t, t), \rho_t^2 I), \tag{2}$$

where $\mu_\theta$ is a neural network that computes the mean of the normal distribution, and typically the term $\rho_t$ is predetermined. A network $\epsilon_\theta$ predicting noise $\epsilon$ can be obtained through parameterization from $\mu_\theta$, i.e. $\mu_\theta = \frac{1}{\sqrt{1-\beta_t}}x_t - \frac{\beta_t}{\sqrt{1-\alpha_t}\sqrt{1-\beta_t}}\epsilon_\theta$, where $\alpha_t = \prod_{i=1}^t 1 - \beta_i$. The model $\epsilon_\theta$ is trained to minimise a simplified variational bound, resulting in the following objective:

$$\mathcal{L}_{DM} = \mathbb{E}_{x,\epsilon\sim\mathcal{N}(0,I),t}\left[\|\epsilon - \epsilon_\theta(x_t, t)\|^2\right]. \tag{3}$$

Latent diffusion models (LDMs) (Rombach et al., 2022) improve efficiency by performing the diffusion process in a lower-dimensional latent space. An autoencoder is first trained, where the encoder $\mathcal{E}_\phi$ maps the input data x to a latent representation $z = \mathcal{E}_\phi(x)$, and the decoder $\mathcal{D}_\delta$ reconstructs the data as $\tilde{x} = \mathcal{D}_\delta(z)$. Diffusion is then applied in the latent space, with the denoising network $\epsilon_\theta$ training objective as:

$$\mathcal{L}_{LDM} = \mathbb{E}_{\mathcal{E}(x),\epsilon\sim\mathcal{N}(0,I),t}\left[\|\epsilon - \epsilon_\theta(z_t, t)\|^2\right]. \tag{4}$$

Xu et al. (2023) extend latent diffusion models to molecular geometries $G = \langle \mathbf{x}, \mathbf{h} \rangle$. The encoding and decoding processes are formulated as $q_\phi(\mathbf{z}_x, \mathbf{z}_h|\mathbf{x}, \mathbf{h}) = \mathcal{N}(\mathcal{E}_\phi(\mathbf{x}, \mathbf{h}), \sigma_0 I)$ and $p_\delta(\mathbf{x}, \mathbf{h}|\mathbf{z}_x, \mathbf{z}_h) = \prod_{i=1}^N p_\delta(x_i, h_i|\mathbf{z}_x, \mathbf{z}_h)$ respectively. To ensure translation invariance, the latent representation $\mathbf{z}_x$ and reconstructed coordinates $\mathbf{x}$ are constrained to the subspace where the centre of mass is zero, i.e., $\sum_i \mathbf{z}_{x,i} = 0$ (or $\sum_i \mathbf{x}_i = 0$). This constraint is also applied to all intermediate diffusion steps. The denoising model is then trained with the following loss:

$$\mathcal{L}_{LDM} = \mathbb{E}_{\mathcal{E}(G),\epsilon\sim\mathcal{N}(0,I),t}\left[\|\epsilon - \epsilon_\theta(\mathbf{z}_{x,t}, \mathbf{z}_{h,t}, t)\|^2\right]. \tag{5}$$

# 4 OUR APPROACH

In this section, we present the proposed `IR-GeoDiff` model. We first train a Transformer-based spectral classifier $\tau_\theta$ to extract spectral and functional group features from the input. Our latent diffusion model $\epsilon_\theta$ maintains roto-translational equivariance throughout the diffusion process. We incorporate IR spectral information into the denoising process using cross-attention mechanisms, enabling spectrum-conditioned molecular recovery. An overview is shown in Figure 1.

## 4.1 SPECTRAL CLASSIFIER AND FUNCTIONAL GROUP REPRESENTATION

Following Wu et al. (2025), we adopt a patch-based embedding layer to extract local spectral features from the IR spectrum and employ a standard Transformer encoder to capture the global spectral characteristics, as shown in Figure 1A. Additionally, we incorporate the chemical formula as input, as previous studies (Alberts et al., 2024a; Wu et al., 2025; Alberts et al., 2025; 2024b) have demonstrated that chemical formulas significantly enhance the model's interpretive capability regarding spectral data. Let $p$ as the number of spectral patches and $c$ as the length of the split chemical formula, the spectral encoder generates the spectral feature $S \in \mathbb{R}^{(p+c)\times d_s}$. To ensure that the spectral encoder can effectively comprehend IR spectra, we implement a classifier to learn the spectral representations by categorising IR spectra according to the functional groups present within the corresponding molecules. We introduce functional group queries as a learnable matrix $M \in \mathbb{R}^{m \times d_{fg}}$, where $m$ is

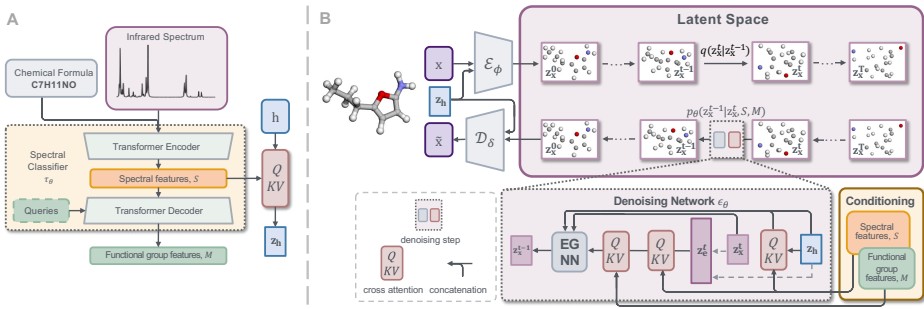

Figure 1: Overview of the proposed `IR-GeoDiff`. A: Spectral features $S$ and functional group features $M$ are extracted by a Transformer-based spectral classifier $\tau_\theta$. B: The encoder $\mathcal{E}_\phi$ maps molecular geometries into a latent representation $\mathbf{z_x}$, which is perturbed through a forward diffusion process and denoised by an equivariant network $\epsilon_\theta$ conditioned on the spectral and functional group features. These features are injected into node and edge representations via cross-attention. Finally, the decoder $\mathcal{D}_\delta$ reconstructs the 3D geometry from the denoised latent representation.

the number of pre-defined functional groups and $d_{\text{fg}}$ is the feature dimensionality. The queries are randomly initialised and fed into the Transformer-based decoder together with the spectral feature $S$. Through cross-attention, $M$ interacts with the spectral representations $S$ to produce functional group embeddings, which are then used for multi-label classification. For each spectrum $S$, the multi-label $\mathbf{y} \in \{0,1\}^m$. The binary cross-entropy (BCE) loss is used to solve this $m$ binary classification problems,

$$\mathcal{L}_{cls} = \frac{1}{m} \sum_{k=1}^{m} [\mathbf{y}_k \cdot \log \tilde{\mathbf{y}}_k + (1 - \mathbf{y}_k) \cdot \log(1 - \tilde{\mathbf{y}}_k)], \tag{6}$$

where $\tilde{\mathbf{y}}_k = \text{sigmoid}(f_S^k)$, and $f_S^k$ is the classifier's output of the $k$th group.

## 4.2 GEOMETRIC AUTO-ENCODING

As we only focus on the perturbed positions, we use the autoencoder to learn a latent space of $\mathbf{x}$. Precisely, given a geometry $G = \langle \mathbf{x}, \mathbf{h} \rangle$, we pre-process $\mathbf{h}$ to obtain latent $\mathbf{z_h}$, and our encoder $\mathcal{E}_\phi$ encodes $\mathbf{x}$ along with $\mathbf{z_h}$, i.e. $\mathbf{z_x} = \mathcal{E}_\phi(\mathbf{x}, \mathbf{z_h})$. The decoder reconstruct the positions from the latent, $\tilde{\mathbf{x}} = \mathcal{D}_\delta(\mathbf{z_x}, \mathbf{z_h})$, thus giving the geometry $\tilde{G} = \langle \tilde{\mathbf{x}}, \mathbf{h} \rangle$. Therefore, the encoding and decoding process can be formulated by $q_\phi(\mathbf{z_x}|\mathbf{x}, \mathbf{z_h}) = \mathcal{N}(\mathcal{E}_\phi(\mathbf{x}, \mathbf{z_h}), \sigma_0 I)$ and $p_\delta(\mathbf{x}|\mathbf{z_x}, \mathbf{z_h}) = \prod_{i=1}^{N} p_\delta(x_i|\mathbf{z_x}, \mathbf{z_h})$ respectively. Following Xu et al. (2023), the encoder $\mathcal{E}_\phi$ and decoder $\mathcal{D}_\delta$ of our autoencoder are based on equivariant graph neural networks (EGNNs) (Satorras et al., 2021) in order to incorporate equivariance into $\mathcal{E}_\phi$ and $\mathcal{D}_\delta$. A brief overview of equivariance is provided in Appendix A, and architectural details of EGNNs are given in Appendix B.1. The autoencoder is optimised by

$$\begin{aligned} \mathcal{L}_{AE} &= \mathcal{L}_{recon} + \mathcal{L}_{reg}, \\ \mathcal{L}_{recon} &= -\mathbb{E}_{q_\phi(\mathbf{z_x}|\mathbf{x}, \mathbf{z_h})} p_\delta(\mathbf{x}|\mathbf{z_x}, \mathbf{z_h}), \end{aligned} \tag{7}$$

where $\mathcal{L}_{reg}$ is KL-reg (Rombach et al., 2022), a slight Kullback-Leibler penalty of $q_\phi$ towards standard Gaussians similar to variational AE, and $\mathcal{L}_{recon}$ in practice is calculated as $L_2$ norm of $\mathbf{x}$ and $\tilde{\mathbf{x}}$.

To obtain the latent atomic representations $\mathbf{z_h}$, we follow the tokenisation approach analogous to that used in natural language processing. Specifically, we first construct a vocabulary of all atom types appearing in the dataset. Each atom type is treated as a discrete token and mapped to a $d_h$-dimensional embedding via a learnable embedding matrix $T \in \mathbb{R}^{n \times d_h}$, where $n$ is the vocabulary size. Given a molecule with $N$ atoms, this results in atomic features $\mathbf{z_h} \in \mathbb{R}^{N \times d_h}$. Spectral information is then introduced to atomic features $\mathbf{z_h}$ by multi-head cross-attention. For attention head $i$, with $Q = W_Q^{(i)} \cdot \mathbf{z_h}, K = W_K^{(i)} \cdot S, V = W_V^{(i)} \cdot S$,

$$\text{Attention}(Q, K, V) = \text{softmax}\left(\frac{QK^T}{\sqrt{d}}\right) \cdot V, \tag{8}$$

where $W_Q^{(i)} \in \mathbb{R}^{d \times d_h}, W_K^{(i)}, W_V^{(i)} \in \mathbb{R}^{d \times d_s}$ are learnable projection matrices.

### 4.3 IR SPECTRA CONDITIONED LATENT DIFFUSION

As IR spectroscopy is derived from the vibration of local substructures of molecules, we introduce spectral information to both node features and edge features.

**Edge construction.** Edge features $\mathbf{z}_e \in \mathbb{R}^{N(N-1) \times d_e}$ are constructed exclusively from invariant quantities. For an edge between atoms $i$ and $j$, the feature $\mathbf{z}_{e_{ij}}$ is generated from the squared distance $\|\mathbf{z}_{\mathbf{x}_i} - \mathbf{z}_{\mathbf{x}_j}\|^2$ and the invariant latent features $\mathbf{z}_{\mathrm{h}_i}, \mathbf{z}_{\mathrm{h}_j}$, i.e. $\mathbf{z}_{e_{ij}} = \mathrm{MLP}(\mathrm{concat}(\|\mathbf{z}_{\mathbf{x}_i} - \mathbf{z}_{\mathbf{x}_j}\|^2, \mathbf{z}_{\mathrm{h}_i} + \mathbf{z}_{\mathrm{h}_j}))$. By construction, this design guarantees that edge features are invariant to both rotation and translation.

To inject spectral information into the denoising process while preserving SE(3)-equivariance, we use the invariant atomic type embeddings $\mathbf{z}_{\mathrm{h}}$ to interact with the spectral features $S$ via cross-attention. The resulting invariant representation $\mathbf{z}_{hs}$ is concatenated with $\mathbf{z}_{\mathrm{h}}$ and passed to the EGNN backbone. We further apply cross-attention to incorporate spectral features $S$ into the edge representations $\mathbf{z}_e$. To capture local connectivity patterns associated with functional groups, the functional group features $M$ obtained from the spectral classifier $\tau_\theta$ are also integrated into $\mathbf{z}_e$ through an additional cross-attention layer.

We adopt EGNNs as the backbone of the denoising network $\epsilon_\theta$, which preserves both invariance and equivariance. Conditioning on IR spectra $S$ and functional groups $M$, we then learn the conditional LDM via

$$\mathcal{L}_{LDM} = \mathbb{E}_{\mathcal{E}(\mathbf{x}), \epsilon \sim \mathcal{N}(0, I), t} \left[ \| \epsilon - \epsilon_\theta(\mathbf{z}_{\mathbf{x}}^t, t, \mathbf{z}_{\mathrm{h}}, S, M) \|^2 \right]. \tag{9}$$

### 4.4 TRAINING AND SAMPLING

**Training.** We pretrain the spectral classifier $\tau_\theta$ using the classification loss $\mathcal{L}_{cls}$. During autoencoder training, the classifier $\tau_\theta$ is jointly optimised with the autoencoder to ensure the alignment with the learned spectral features. The overall objective in this stage is $\mathcal{L}_{AE} + \mathcal{L}_{cls}$. In the subsequent diffusion training stage, the spectral classifier is frozen to ensure a stable and consistent conditioning signal, while the autoencoder remains learnable. The model is optimized with $\mathcal{L}_{AE} + \mathcal{L}_{LDM}$.

**Sampling.** Given the number $N$ and the types $\mathbf{h}$ of atoms, we first sample a latent code $\mathbf{z}_{\mathbf{x}}^T$ from the standard normal distribution. This latent is then denoised iteratively by using $\mathbf{z}_{\mathbf{x}}^t$ by $\mathbf{z}_{\mathbf{x}}^{t-1} = \frac{1}{\sqrt{1-\beta_t}}(\mathbf{z}_{\mathbf{x}}^t - \frac{\beta_t}{\sqrt{1-\alpha_t}} \epsilon_\theta(\mathbf{z}_{\mathbf{x}}^t, t, \mathbf{z}_{\mathrm{h}}, S, M)) + \rho_t \epsilon$. After the denoising process, the latent $\mathbf{z}_{\mathbf{x}}$ is decoded into the molecular geometry via $p_\delta(\mathbf{x}|\mathbf{z}_{\mathbf{x}}, \mathbf{z}_{\mathrm{h}})$, resulting in a final geometry $G = \langle \mathbf{x}, \mathbf{h} \rangle$.

### 4.5 EVALUATION METRICS FOR CONDITIONAL RESULTS

We evaluate conditioning performance from two perspectives: spectral similarity and structural similarity. As our goal is to recover molecular structures consistent with a given IR spectrum, we expect the sampled molecules to exhibit both high spectral similarity to the input spectrum and high structural similarity to the reference molecule.

**Spectral similarity.** To measure spectral similarity, we adopt the Spectral Information Similarity (SIS) metric $\mathrm{SIS}(S_G, S)$ proposed by McGill et al. (2021), which quantifies the similarity between the given spectrum $S$ and the spectra of the sampled molecule $S_G$. Further details are provided in Appendix E.5. SIS also serves as a physically grounded measure of geometric correctness, since IR spectra are computed directly from the full three-dimensional geometry. In typical IR spectral analysis, chemists primarily focus on the functional group region (1,500–4,000 cm$^{-1}$), where distinct peaks corresponding to specific functional groups or chemical bonds are found. The fingerprint region (400–1,500 cm$^{-1}$) is typically used for final confirmation of molecular identity (Clayden et al., 2012). We further define SIS$^*$, which computes the SIS score restricted to the functional group region.

**Structural similarity.** Due to the theoretically one-to-one correspondence between molecular structures and their IR spectra (ignoring enantiomers), we also evaluate the structural similarity between sampled molecules and the reference structures. After sampling geometries $G = \langle \mathbf{x}, \mathbf{h} \rangle$, we further convert them into molecular graph using a valence-guided bond assignment algorithm, as detailed in Appendix D.1. Following Adams & Coley (2023) and Chen et al. (2023), we measure chemical graph similarity $\mathrm{sim}_g(\tilde{G}, G)$, defined as the Tanimoto similarity between Morgan fingerprints computed using RDKit (Landrum et al.). Inspired by the top-$n$ accuracy metrics used in prior works (Alberts et al., 2024a; Wu et al., 2025; Alberts et al., 2025) we further define *molecular accuracy* to evaluate

Table 1: Comparison with baseline models and ablation variants on chemical graph similarity ($\text{sim}_g$), molecular accuracy, Spectral Information Similarity (SIS), and SIS*. Due to the high computational cost of quantum chemical calculations with Gaussian16, SIS and SIS* are computed on a subset of 200 test spectra, each with 50 sampled molecules (appr. 10,000 spectra in total). See Appendix C for details. *fg ca*: cross-attention between edge features and functional group features; *atom ca*: cross-attention between atomic node features and IR spectral features.

| Method | $\text{sim}_g$ | $\max \text{sim}_g$ | mol acc(%) | SIS | max SIS | SIS* |
|---|---|---|---|---|---|---|
| EDM (Hoogeboom et al., 2022) | 0.194 | 0.448 | 2.73 | 0.351 | 0.578 | 0.355 |
| GEOLDM (Xu et al., 2022)) | 0.227 | 0.567 | 11.9 | 0.380 | 0.649 | 0.390 |
| ours w/o edge | 0.478 | 0.592 | 28.0 | 0.502 | 0.623 | 0.524 |
| ours w/o atom ca | 0.541 | 0.954 | 85.7 | 0.591 | 0.896 | 0.633 |
| ours w/o fg ca | 0.660 | 0.965 | 89.5 | 0.639 | 0.930 | 0.672 |
| **ours** | **0.672** | **0.969** | **90.6** | **0.644** | **0.931** | **0.681** |

the model's overall performance across the test set. For each input IR spectrum, we generate multiple molecular samples; if at least one sampled molecule exactly matches the reference structure, the test case is considered successful. Molecular accuracy is computed as the proportion of successful test cases over the entire test set.

## 5 EXPERIMENTS

**Datasets.** We use the QM9S dataset (Zou et al., 2023), which is derived from the QM9 dataset (Ramakrishnan et al., 2014) and contains approximately 130k small molecules composed of five atom types (H, C, N, O, and F), with up to nine heavy atoms. QM9S extends QM9 by providing multiple types of spectra. In particular, IR spectra are obtained by re-optimising molecular geometries and performing vibrational analysis using quantum chemical calculations with Gaussian16 (Frisch et al., 2016). Prior to training, we perform a data screening step to remove invalid molecules. Following Ayadi et al. (2024); Chen et al. (2023), we shuffle all samples and then randomly select 1,000 molecules as the test set. The remaining molecules are split into training and validation sets with a 95:5 ratio. Further details are provided in Appendix D.

In this section, we report results based on the proposed metrics that evaluate how well the sampled molecules match the input IR spectra. Spectra of sampled molecules are computed using the same procedure as in QM9S (details in Appendix C). All evaluations are performed on stable molecules only, as unstable samples often undergo significant structural changes during geometry optimisation with Gaussian16 (Frisch et al., 2016), resulting in spectra that no longer reflect the original structures. Structural quality metrics (e.g. validity, stability, and connectivity) are reported in Appendix E.4.

### 5.1 RECOVERY OF MOLECULAR GEOMETRIES FROM IR SPECTRA

**Baselines.** We compare `IR-GeoDiff` against two representative 3D molecular generative models: EDM (Hoogeboom et al., 2022) and GEOLDM (Xu et al., 2022). EDM operates directly in atomic coordinate space, while GEOLDM performs generation in a learned latent space. We follow their conditioning mechanisms by concatenating conditioning features to the node representations.

For each test IR spectrum, we sample 50 molecules, and the results are reported in Table 1. Additional results with varying numbers of samples are provided in Appendix E.3. As shown in Table 1, `IR-GeoDiff` achieves a molecular accuracy of 90.6%, demonstrating its ability to accurately recover the distribution of molecular geometries consistent with the input IR spectrum. Compared to GEOLDM, our model improves SIS by 0.264 and SIS* by 0.291, with the larger gain on SIS* highlighting its stronger performance in the functional group region, which is a more structurally informative part of the spectrum. Some sampling results are illustrated in Figure 2.

Across all metrics, our model consistently outperforms both baselines. A key reason for the performance gap is that baseline models are designed to generate diverse molecules, even under conditional settings, and do not constrain atom types or counts during generation. In contrast, our IR-GeoDiff aims to *recover a precise molecular distribution defined by the given IR spectrum*. Thus, both atom types and counts are provided as part of the input to reduce the search space. For a fair comparison, we modify the baseline sampling procedures by fixing the number of atoms. Atom types remain un-

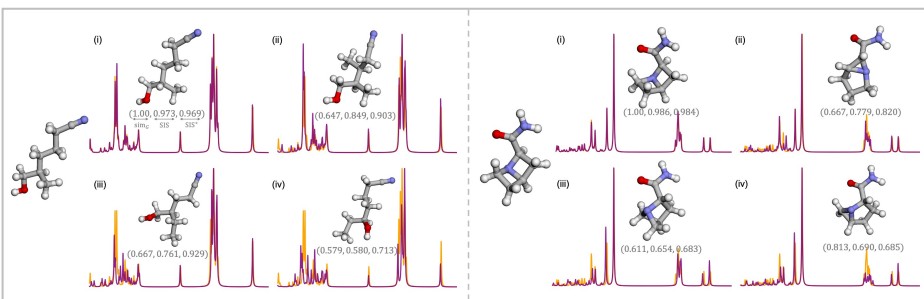

Figure 2: Examples of molecules sampled under IR spectral conditions. For spectra, orange curves denote input IR spectra; purple curves denote spectra of sampled molecules. For atoms, hydrogen: white, carbon: gray, oxygen: red, nitrogen: blue.

constrained in the baselines because their denoising networks jointly model position and feature-level noise, which does not support atom types being specified as input.

**Ablation study.** We perform ablation studies to assess the contributions of key components in our denoising network, including: (1) the construction of edge features, (2) the cross-attention between atomic node features and IR spectral features, and (3) the cross-attention between edge features and functional group features. As shown in Table 1, removing any of these modules leads to a performance drop across both spectral and structural metrics. Notably, removing edge features leads to a substantial drop in molecular accuracy. This is because the spectral information modulates only the invariant node features $z_h$, which influence the coordinate predictions $z_x$ only indirectly through the EGNN layers. These results highlight that edge-level information is crucial for aligning structural generation with spectral signals. This observation is also consistent with the physical principles of IR spectroscopy, where absorption peaks primarily reflect bond vibrations rather than atomic identities alone. Overall, these results suggest that each module contributes to reducing ambiguity in the inverse mapping from spectrum to structure, allowing the model to better recover the correct molecular distribution.

## 5.2 UNDERSTANDING SPECTRAL-STRUCTURAL RELATIONSHIPS VIA ATTENTION VISUALISATION

We visualise the normalised attention maps from the cross-attention layers between spectral patch features and edge features, as shown in Figure 3. For each spectral feature, we take the maximum attention score across all edges to identify the spectral regions the model attends to most strongly. Interestingly, we find that the spectral peaks receiving the highest attention often correspond to characteristic functional groups.

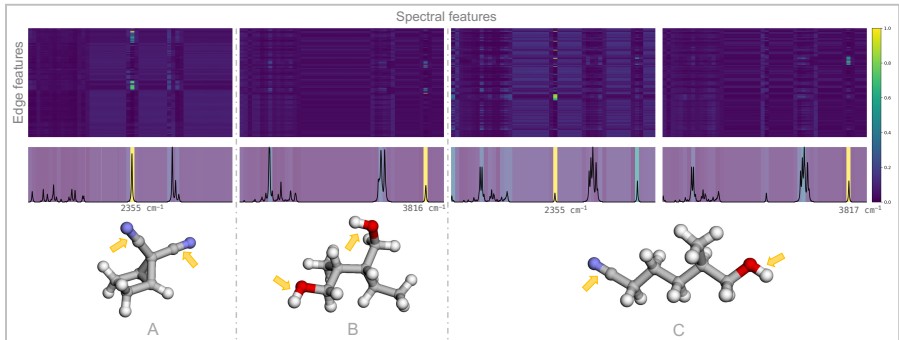

Figure 3: Visualisation of cross-attention between spectral features and edge features. **Top**: normalized attention maps (edge features × spectral features). **Middle**: corresponding IR spectra, where each spectral patch is colored by its maximum attention score across all edges. **Bottom**: molecular geometries with the corresponding functional groups highlighted in yellow.

## 5.3 LIMITATIONS

Figure 3 shows two representative vibrational modes: the stretching vibration of the O−H bond in a hydroxy group and the stretching vibration of the carbon–nitrogen triple bond (C≡N). These correspond to absorption peaks typically observed around 3600 and 2250 cm⁻¹ respectively (Clayden et al., 2012). As all IR spectra are computed using Gaussian16 (Frisch et al., 2016),we further examine the vibrational mode outputs to confirm that the attended peaks indeed arise from the expected functional groups (see Appendix F). In examples A and B, each molecule contains only one type of functional group, and the model correctly identifies the corresponding characteristic spectral peak. In example C, the molecule contains two different functional groups, and we observe that they are captured separately by different cross-attention layers, which attend to the respective spectral peaks associated with each group. This demonstrates its ability to disentangle and localise multiple spectral signatures within a single molecule.

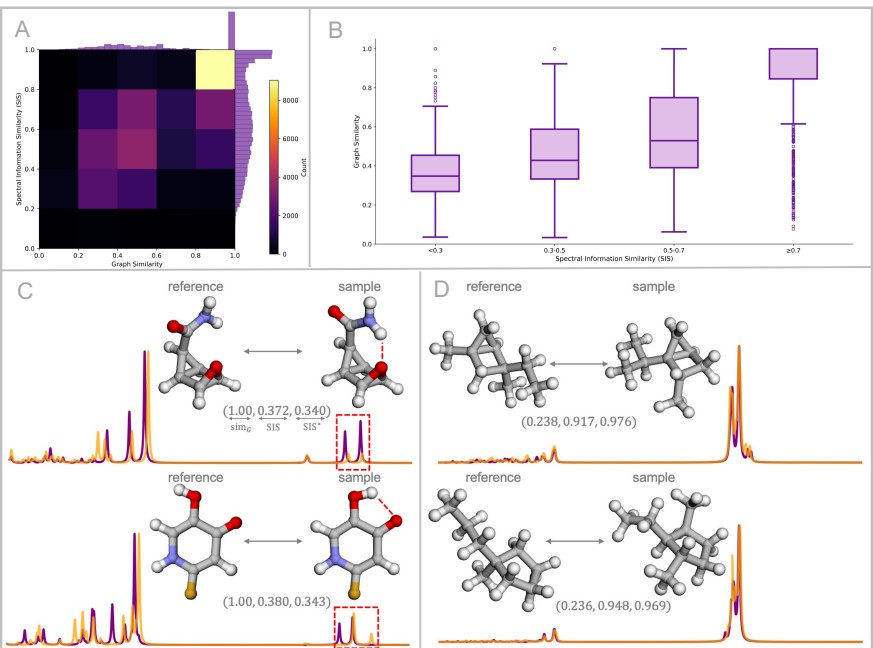

Figure 4: **A:** 2D histogram showing the joint distribution of $\text{sim}_g$ and SIS, computed over 1,000 test spectra, with 50 sampled molecules per spectrum. **B:** Box plots of $\text{sim}_g$ grouped by SIS ranges. **C:** Examples with high $\text{sim}_g$ but low SIS. **D:** Examples of molecules with high SIS but low $\text{sim}_g$.
For spectra, orange curves denote input IR spectra; purple curves denote spectra of sampled molecules.
For atoms, hydrogen: white, carbon: gray, oxygen: red, nitrogen: blue, fluorine: yellow.
Figure 4A presents the joint distribution of graph similarity $\text{sim}_g$ and SIS across 1,000 test spectra. Many samples lie in the mid-range for both metrics, while a sharp density peak appears near the top-right corner, indicating a cluster of molecules that are near-exact matches to the reference. As shown in Figure 2, both metrics are highly sensitive to molecular configuration: even minor changes in atomic connectivity can lead to large drops in both spectral and structural similarity, making perfect matches rare and distinct.

In practical applications, chemists typically identify compounds by visually comparing the IR spectra of unknown samples with those of known references. Motivated by this, we analyse the distribution of graph similarity across different SIS ranges, as shown in Figure 4B. While graph similarity generally increases with SIS, the correlation between the two is moderate. This observation underscores the importance of evaluating both structural and spectral similarity simultaneously, as neither metric alone is sufficient to fully characterize the correspondence between sampled and reference molecules.

Notably, there exists a non-negligible number of cases where molecules exhibit high graph similarity but low SIS, and vice versa. As shown in Figure 4C, samples with high $\text{sim}_g$ but low SIS, primarily caused by conformational changes that result in the formation of intramolecular hydrogen bonds. Although hydrogen bonds are not conventional covalent bonds, they represent relatively strong

electrostatic interactions, typically between hydrogen atoms and electronegative atoms such as N, O, or F atoms. These interactions can shift the vibrational frequencies of associated functional groups, leading to notable discrepancies in the IR spectra despite high structural similarity (Clayden et al., 2012; Freedman, 1961). This suggests that while our model currently has limited control over molecular conformation, it also emphasises the importance of interpreting IR spectra within the full three-dimensional geometric context. Samples with low $sim_g$ but high SIS are shown in Figure 4D. This typically arises from mismatches in the molecular scaffolds, particularly when molecules lack distinctive functional groups and consist of carbon and hydrogen atoms. In such cases, the IR spectral signals reflecting differences in carbon backbone topology are often subtle and difficult to interpret, highlighting the limited ability of IR spectroscopy to resolve differences in molecular skeletons. Future work could incorporate additional spectral modalities, particularly nuclear magnetic resonance (NMR), which offers complementary information. For instance, $^1$H-NMR and $^{13}$C-NMR spectra are highly informative about molecular backbones and can also reveal conformational details (Vakili et al., 2012; Tormena, 2016). While our model already achieves strong performance using IR spectra alone, incorporating NMR could further improve accuracy and impose stronger constraints on 3D molecular recovery.

## 6 CONCLUSION

In this paper, we presented a new approach to recover the distribution of three-dimensional molecular geometries from a given infrared (IR) spectrum. To this end, we propose `IR-GeoDiff` the first model that directly generates 3D molecular structures from IR spectra. Our experiments demonstrated that IR-GeoDiff effectively captures the underlying distribution, with interpretability analyses showing strong alignment with quantum mechanical principles of IR spectroscopy. By examining cases with high graph similarity but low SIS, and vice versa, we observe that the current model still has limited control over molecular conformations. In addition, IR spectra contain intrinsic ambiguity for distinguishing certain molecular scaffolds. This motivates the use of additional spectral modalities such as NMR to provide complementary structural information and stronger constraints on 3D recovery in the future work.

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

## APPENDIX

## CONTENT

In this appendix, we provide additional information to support the main text. Specifically:

- In Section A, we provide a brief overview of the concept of equivariance, which is essential for modelling molecular geometries.

- Section B presents the architecture details of our model, with the Equivariant Graph Neural Networks (EGNNs) and the implementation details of each component.

- In Section C, we describe the procedure used to compute IR spectra for sampled molecules using the Gaussian16 package.

- Section D introduces our method for converting 3D coordinates into molecular graph, and we use it to screen and filter mismatched data in the QM9S dataset and also present the molecular size distributions of the datasets.

- In Section E, additional experimental results are reported, including the performance of our spectral classifier, the results of IR-GeoDiff across multiple sampling runs and different sampling counts, and standard structure quality metrics including validity, stability, and connectivity.

- Section F provides sample outputs from vibrational analysis conducted using Gaussian16 package.

- In Section G, we describe our limited use of Large Language Models (LLMs) for polishing the writing of this manuscript.

## A  EQUIVARIANCE

For geometric systems like molecules, it is essential to ensure equivariance, as directional features such as atomic forces should transform consistently with changes in molecular coordinates (Thomas et al., 2018; Weiler et al., 2018; Fuchs et al., 2020; Batzner et al., 2022). Let $T_g : X \to X$ and $S_g : Y \to Y$ be two sets of transformation on spaces $X$ and $Y$ respectively for an abstract group element $g \in G$. A function $\phi : X \to Y$ is equivariant to $g$ if

$$\phi(T_g(\mathbf{x})) = S_g(\phi(\mathbf{x})). \tag{10}$$

For molecular geometries, denote $\mathbf{x} = (x_1, \ldots, x_N) \in \mathbb{R}^{N \times d}$ as an input $N$ point clouds embedded in a $d$-dimensional space, and $\mathbf{y} = \phi(\mathbf{x})$ as the transformed set of point clouds with a non-linear function $\phi(\cdot)$. In this work, we adopt the SE(3)-equivariant latent space formulation proposed by Xu et al. (2023), where both the theoretical definition and practical implementation of equivariance are derived. Specifically, we consider the Special Euclidean group SE(3), i.e. the group of rotation and translation in 3D space, where transformations $T_g$ and $S_g$ can be represented by a translation $t$ and an orthogonal matrix rotation $R$.

For translation equivariance, translating the input by $g \in \mathbb{R}^d$ results in an equivalent translation of the output, $\phi(x+g) = \phi(x)+g$, where $x+g$ denotes the translated point cloud $(x_1+g, \ldots, x_N+g)$. For rotation equivariance, rotating the input by an orthogonal matrix $Q \in \mathbb{R}^{d \to d}$ yields a correspondingly rotated output $\phi(Qx) = Q\phi(x)$ where $Qx = (Qx_1, \ldots, Qx_N)$.

## B  ADDITIONAL DETAILS ON EXPERIMENTS

### B.1  E(N)-EQUIVARIANT GRAPH NEURAL NETWORKS (EGNNS)

Neural networks used for the autoencoder and the backbone of our denoise network are implemented using EGNNs (Satorras et al., 2021). Denote a molecular graph $\mathcal{G} = (\mathcal{V}, \mathcal{E})$ with nodes $v_i \in \mathcal{V}$ and edges $e_{ij} \in \mathcal{E}$. Each node $v_i$ is associated with a node embeddings $\mathbf{h}_i \in \mathbb{R}^{d_h}$ and a coordinate $\mathbf{x}_i \in \mathbb{R}^{d_x}$ are considered. EGNNs are composed of Equivariant Graph Convolutional Layers (EGCLs).

At layer $l$, the features and coordinates are updated as $\mathbf{h}^{l+1}, \mathbf{x}^{l+1} = EGCL[\mathbf{h}^l, \mathbf{x}^l, \mathcal{E}]$ with following update rules:

$$\mathbf{m}_{ij} = \phi_e\left(\mathbf{h}_i^l, \mathbf{h}_j^l, d_{ij}^2, a_{ij}\right), \tag{11}$$

$$\mathbf{h}_i^{l+1} = \phi_h(\mathbf{h}_i^l, \sum_{j \neq i} \tilde{e}_{ij}\mathbf{m}_{ij}), \tag{12}$$

$$\mathbf{x}_i^{l+1} = \mathbf{x}_i^l + \sum_{j \neq i} \frac{\mathbf{x}_i^l - \mathbf{x}_j^l}{d_{ij} + 1} \phi_x(\mathbf{h}_i^l, \mathbf{h}_j^l, d_{ij}^2, a_{ij}) \tag{13}$$

where $d_{ij}^2 = \|\mathbf{x}_i^l - \mathbf{x}_j^l\|^2$ is the squared pairwise distance and $a_{ij}$ is the edge attribute between atoms $v_i$ and $v_j$. $\tilde{e}_{ij} = \phi_{inf}(\mathbf{m}_{ij})$ serves as the attention weights to reweight messages passed from different edges. Following prior work (Satorras et al., 2021; Hoogeboom et al., 2022; Xu et al., 2023), $\mathbf{x}_i^l - \mathbf{x}_j^l$ is normalized by $d_{ij} + 1$. $\phi_e, \phi_h, \phi_x$ and $\phi_{inf}$ are implemented as Multi Layer Perceptrons (MLPs). EGNNs maintain equivariance to both rotations and translations of the input coordinates $\mathbf{x}_i$, and are permutation-equivariant with respect to node orderings, consistent with standard graph neural networks (GNNs).

## B.2 MODEL ARCHITECTURE DETAILS

**Spectral classifier $\tau_\theta$.** The spectral classifier is implemented as a Transformer (Vaswani et al., 2017) with 4 encoder and 4 decoder layers when trained on the QM9S dataset. Both encoder and decoder use 8-head multi-head attention with a hidden dimensionality of 512. Following Wu et al. (2025), we employ a patch-based spectral embedding layer to process the IR spectra: 3,200 points are uniformly sampled with a patch size of 64, resulting in 50 spectral patches.

**Autoencoder.** The encoder $\mathcal{E}_\phi$ and decoder $\mathcal{D}_\delta$ are implemented using EGNNs (Satorras et al., 2021), as described in Appendix B.1. Following the design in GeoLDM (Xu et al., 2023), we use a 1-layer EGNN for the encoder and a 9-layer EGNN with 256 hidden dimensions for the decoder. The dimensionality of the latent atom embedding $\mathbf{z}_\mathrm{h}$ is 16, and the coordinate latent $\mathbf{z}_\mathrm{x}$ has dimensionality 3.

**Denoising network $\epsilon_\theta$.** The denoising network is also implemented as a 9-layer EGNN with 256 hidden dimensions. Spectra-to-node and spectra-to-edge cross-attention modules use 4 layers each, while functional group-to-edge cross-attention uses 2 layers. The dimensionality of the edge features is set to 16.

All models use SiLU activation functions and are implemented using the PyTorch framework (Paszke et al., 2019).

## B.3 DETAILS OF TRAINING AND SAMPLING

We train all modules until convergence using the AdamW optimizer (Loshchilov & Hutter, 2019). Following the learning rate schedule from (Vaswani et al., 2017), the learning rate is varied according to:

$$\text{rate} = 512^{-0.5} \cdot \min(\text{step}^{-0.5}, \text{step} \cdot \text{warmup\_step}^{-1.5}), \tag{14}$$

with 3000 warm-up steps.

The spectral classifier is pretrained for 100 epochs with a base learning rate of 0.8 and a batch size of 256, requiring approximately 3.5 hours. The autoencoder and diffusion model are trained with a base learning rate of 0.1. The autoencoder is trained for 300 epochs with a batch size of 128 (about 20 hours), and the diffusion model is trained for 1,000 epochs with a batch size of 64 (about 163 hours). Sampling 50 molecules for each of the 1,000 test spectra takes approximately 27 hours with a batch size of 128.

For the results of EDM and GEOLDM reported in Table 1, spectral features are obtained from our pretrained spectral classifier. To make the features compatible with their architectures, we use an MLP to project the spectral features $S \in \mathbb{R}^{(p+c) \times d_s}$ to $\mathbb{R}^{1 \times (p+c)}$, allowing it to be concatenated with atomic features. Both EDM and GEOLDM are trained until convergence. Specifically, EDM

is trained for 1100 epochs, while GEOLDM is fine-tuned for 500 epochs starting from the official checkpoint provided by the authors.

All experiments are conducted on a single NVIDIA A100 GPU.

## C    COMPUTATION OF IR SPECTRA

We use the Gaussian16 package (Frisch et al., 2016) to re-optimise the molecular geometries and perform vibrational analysis of the sampled molecules at the B3LYP/def2-TZVP level of theory, consistent with the QM9S dataset (Zou et al., 2023). The same method is also used to broaden the resulting IR spectra. After applying a scaling factor of 0.965 to the computed frequencies, the IR spectra are generated using Lorentzian broadening:

$$Lo(x) = \frac{F}{2\pi} \times \frac{1}{(x - x_n)^2 + 0.25 \times F^2} \times y_n \qquad (15)$$

where $F$ represents the half-width of the peak, set to 15 cm$^{-1}$ for the infrared spectra. $x_n$ and $y_n$ represents the calculated wavenumber and the IR intensity of the $n^{th}$ vibration mode respectively, and $x$ is any wavenumber of the IR spectra.

All quantum chemical calculations were performed using Gaussian16 (Frisch et al., 2016), with 32 GB of allocated memory (%Mem=32GB) and 16 shared CPU cores (%NProcShared=16). The maximum allowed computation time per job was set to 1 hour. On average, evaluating the IR spectrum of one sampled molecule takes approximately 10 minutes. As each sampling round generates 50 molecules for each of the 1,000 test IR spectra (i.e. 50,000 molecules per round), computing IR spectra for all samples is computationally expensive. Therefore, we first performed a one-time full-spectrum evaluation over all samples to assess the stability of the average SIS score on the QM9S test set. As shown in Figure A1, the mean SIS value stabilises once the number of evaluated test spectra exceeds 200. Based on this observation, all subsequent spectral evaluations in our experiments were conducted using a subset of 200 test spectra.

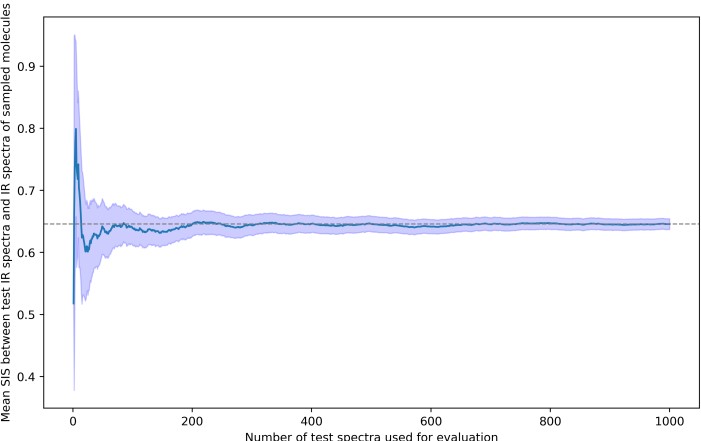

Figure A1: Convergence of the mean Spectral Information Similarity (SIS) as the number of test IR spectra increases. The shaded region indicates the 95% confidence interval of the mean SIS. The curve stabilises after approximately 200 spectra, supporting the use of this subset for efficient and reliable evaluation. The result is based on one full sampling run (50,000 molecules in total).

## D    DATASET PREPROCESSING AND DISTRIBUTION

In this work, we explicitly include hydrogen atoms when reconstructing molecular structures, as vibrations involving hydrogens contribute significantly to IR spectral features. Since we focus

exclusively on neutral molecules without formal charges or radicals, chemical bonds and their orders are inferred based on atomic valence and connectivity rules.

Although the latest version of RDKit (Landrum et al.) provides an internal function, `xyz2mol`, for converting 3D atomic coordinates into RDKit Mol objects, it is also designed to support charged species. As a result, it occasionally fails to produce valid molecular graphs, even for geometry-optimised structures computed using Gaussian16.

### D.1    BOND ASSIGNMENT PROCEDURE

We assign chemical bonds and bond orders step-by-step based on atomic distances and valence constraints. Our method supports complete neutral molecules composed of atoms from the set: H, B, C, N, O, F, Si, P, S, Cl, As, Se, Br.

1. **Construct initial adjacency matrix.** We construct an initial adjacency matrix by assigning a bond between atoms $i$ and $j$ if their distance $d_{ij}$ is less than a threshold $L_{ij} + \delta$, where $L_{ij}$ is the reference single bond length for that atom pair (Hoogeboom et al., 2022), and $\delta = 40$ pm is a distance tolerance. This yields a basic single-bond connectivity graph, which is then converted into an RDKit Mol object.

2. **Update terminal atoms.** After constructing the initial adjacency matrix using single bond thresholds, we identify saturated atoms as those whose valence is fully occupied by current connections. This initially includes hydrogen and halogen atoms, which form only one bond. Starting from the molecular periphery, we iteratively mark all saturated atoms and then identify terminal atoms based on their local environment. A carbon, nitrogen, or oxygen atom is considered terminal if it is connected to only one neighbour that is not yet saturated. For sulfur atoms, we treat them as terminal when they are involved in only one bond. Once all terminal atoms are identified, we increase bond orders along their existing bonds until their valences are satisfied.

3. **Update aromatic rings.** We identify all independent rings in the molecule by using RDKit and check whether they satisfy Hückel's rule (i.e. containing $4n + 2\ \pi$ electrons). For rings deemed aromatic, we assign appropriate bond orders.

4. **Update C and N atoms.** Carbon and nitrogen are capable of forming multiple bonds, including triple bonds, and thus require additional handling. We prioritise atoms with higher unsaturation, and for each, we first attempt to increase bond orders with unsaturated neighbours. If all neighbours are saturated, we consider bonding to atoms such as sulfur or phosphorus if their valence allows. When multiple candidate bonds are equally unsaturated, we prefer shorter bonds, as higher bond orders generally correspond to shorter bond lengths.

5. **Update other atoms.** For the remaining atoms, we follow a strategy similar to `xyz2mol`, iteratively increasing bond orders until all atoms are saturated or no further updates are possible.

6. If no further bond order updates can be made, we attempt to remove the existing bond with the largest deviation from the corresponding standard bond length. Bonds are only removed if the deviation exceeds a threshold of 15 pm. After removal, the bond assignment process restarts from step two. If no further bonds can be removed or the maximum number of iterations is reached, the algorithm returns the final geometry obtained at that point.

### D.2    SCREENING OUT INVALID GEOMETRIES

We apply our bond reconstruction method described in Section D.1 to the molecular geometries provided in the QM9S dataset and identify 603 molecules with inconsistencies between the 3D coordinates and the corresponding SMILES strings. Examples are shown in Figure A2.

### D.3    DATASET DISTRIBUTION

Figure A3 shows the distributions of molecular sizes, measured by the number of atoms. We observe that the test sets of QM9S exhibit distributions that are statistically similar to the full datasets, indicating that the randomly split test set is representative.

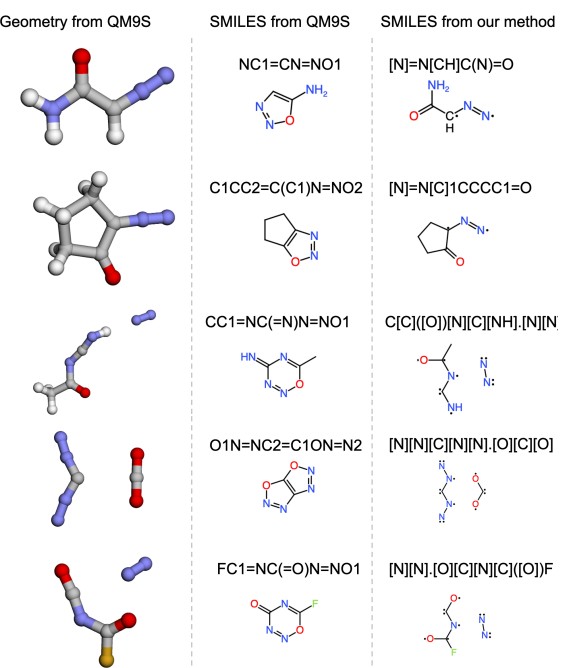

Figure A2: Examples of molecular geometries (hydrogen: white, carbon: gray, oxygen: red, nitrogen: blue, fluorine: yellow) from the QM9S dataset that are inconsistent with their provided SMILES strings. The **left** column shows 3D geometries, the **middle** column displays the corresponding SMILES from QM9S, and the **right** column shows the SMILES reconstructed by our method based on the 3D coordinates. All SMILES reconstructed by our method are consistent with the corresponding molecular geometries. (Molecular graphs converted from SMILES representations are provided, below each SMILES, for easier comparison.)

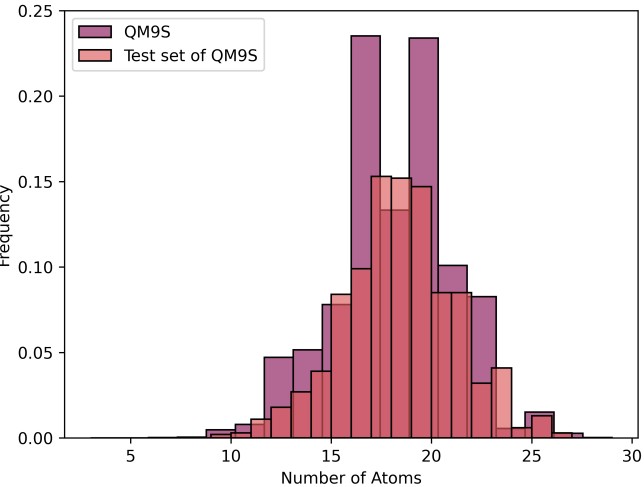

Figure A3: Distribution of molecular sizes (number of atoms).

# E  ADDITIONAL EXPERIMENTAL RESULTS

## E.1  LABELLING OF FUNCTIONAL GROUPS

We use SMARTS strings to identify function groups within molecules, and define 20 types of common functional groups in molecules of QM9S dataset, as shown in Table A1.

Table A1: 20 SMARTS representations of functional groups and their occurrence in QM9S dataset and classification performance of our spectral classifier evaluated by Accuracy and F-1 Score.

| Functional groups | SMARTS | Count | Accuracy | F-1 |
|---|---|---|---|---|
| alkane | [CX4;H3,H2,H1] | 123361 | 1.0000 | 1.0000 |
| alkene | [CX3]=[CX3] | 16382 | 0.9940 | 0.9767 |
| alkyne | [CX2]#[CX2] | 16976 | 0.9980 | 0.9917 |
| amine | [NX3;!$(NC=O)] | 38933 | 0.9980 | 0.9966 |
| imine | [NX2]=[CX3] | 15683 | 0.9980 | 0.9931 |
| nitrile | [NX1]#[CX2] | 16356 | 0.9980 | 0.9896 |
| alcohol | [OX2H;!$(OC=O)] | 42589 | 0.9990 | 0.9985 |
| ether | [OX2H0;!$(OC=O);!$([O]-[O])] | 51295 | 0.9950 | 0.9937 |
| haloalkane | [#6;!$(C(=O)[F])][F] | 2054 | 1.0000 | 1.0000 |
| aldehyde | [#6,H][CX3H1](=O) | 14828 | 0.9970 | 0.9861 |
| ketone | [#6][CX3](=O)[#6] | 14690 | 0.9950 | 0.9772 |
| ester | [CX3](=O)[OX2H0] | 7011 | 0.9940 | 0.9471 |
| amide | [CX3](=O)[NX3] | 13541 | 0.9980 | 0.9918 |
| arene | [$([cX2](:*):*),$([cX3](:*):*)] | 21687 | 0.9980 | 0.9947 |
| imidazole | [#7]:[#6]:[#7] | 6486 | 0.9950 | 0.9331 |
| pyrazole | [#7]:[#7] | 6927 | 0.9980 | 0.9823 |
| oxazole | [#7]:[#6]:[#8] | 3706 | 0.9980 | 0.9613 |
| isoxazole | [#7]:[#8] | 3811 | 0.9980 | 0.9652 |
| cyclopropane | C1CC1 | 30386 | 0.9860 | 0.9713 |
| epoxide | C1OC1 | 10200 | 0.9930 | 0.9540 |

## E.2  EVALUATION ACROSS MULTIPLE SAMPLING RUNS

We conduct repeated sampling experiments and report the variability in performance across three independent runs. Table A2 summarises the mean and standard deviation of key evaluation metrics, including chemical graph similarity, spectral similarity (SIS and SIS$^*$), and molecular accuracy. All results are computed on stable sampled molecules, as described in Section 5.

Table A2: Evaluation of IR-GeoDiff trained on QM9S dataset over three repeated sampling runs. We report the mean and standard deviation for chemical graph similarity ($\mathrm{sim}_g$), Spectral Information Similarity (SIS), SIS$^*$, and molecular accuracy.

| Method | $\mathrm{sim}_g$ | $\max \mathrm{sim}_g$ | Mol acc(%) | SIS | max SIS | SIS$^*$ |
|---|---|---|---|---|---|---|
| ours | 0.672±0.001 | 0.969±0.002 | 90.6±0.8 | 0.644±0.003 | 0.931± 0.002 | 0.681± 0.004 |

## E.3  EVALUATION UNDER DIFFERENT SAMPLING COUNTS

We analyse the effect of varying the number of samples per input spectrum (1, 5, 10, 20, 30, 40, 50) on the QM9S dataset. As shown in Figure A4, the graph similarity $\mathrm{sim}_g$ remains stable across different sample sizes, whereas both $\max \mathrm{sim}_g$ and molecular accuracy consistently improve as the number of samples increases. The performance gains begin to saturate beyond 30 samples, indicating that using 50 samples provides a stable and representative evaluation of the model's ability to recover the correct structure.

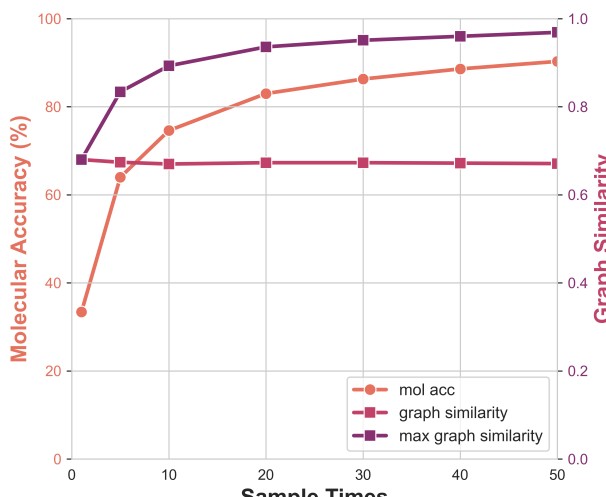

Figure A4: Effect of the number of samples per input spectrum on model performance, reporting molecular accuracy (mol acc), average graph similarity ($\text{sim}_g$), and maximum graph similarity ($\max \text{sim}_g$) across different sampling times.

### E.4 VALIDITY, STABILITY AND CONNECTIVITY

We evaluate the quality of generated molecules using three common structural metrics: validity, stability, and connectivity. We define validity as the proportion of generated molecules in which no atom exceeds its maximum allowed valence, and the overall molecular graph is connected, given that the atom types and counts are predefined. Stability measures the proportion of molecules where all atoms are fully saturated up to their permitted valence limits. Connectivity measures the proportion of molecules whose corresponding graphs are connected. We report these metrics in Table A3 for our full model and baseline models trained on the QM9S dataset for reference, to provide a general assessment of molecular quality in the context of generative modelling. *But please note, these are not the primary evaluation metrics for our geometry recovery task, and are inappropriate to evaluate our approach.* Our main focus is on spectral and structural fidelity, as measured by SIS and $\text{sim}_g$ in Table 1 in the main paper, respectively.

Table A3: Comparison with baseline models on validity, stability, and connectivity.

| Method | Validity(%) | Stability(%) | Connectivity(%) |
|---|---|---|---|
| EDM (Hoogeboom et al., 2022) | 99.5 | 73.6 | 98.1 |
| GEOLDM (Xu et al., 2023) | 99.7 | 95.0 | 99.5 |
| ours | 96.9±0.03 | 82.5±0.07 | 95.8±0.06 |

### E.5 SPECTRAL INFORMATION SIMILARITY

The spectral information similarity (SIS) proposed by McGill et al. (2021) are calculated to compare the spectra of sampled molecules and the input spectra. First, each spectrum was normalized by summing all absorbance values to unity. We then calculated the Spectral Information Divergence (SID), which measures the divergence and peak overlap between any two spectra. Finally, SIS was computed based on SID as follows:

$$\text{SID}(\breve{Y}_{\text{sampled}}, \breve{Y}_{\text{input}}) = \sum_i y_{\text{sampled,i}} \ln \frac{y_{\text{sampled, i}}}{y_{\text{input,i}}} + y_{\text{input,i}} \ln \frac{y_{\text{input,i}}}{y_{\text{sampled,i}}}, \quad (16)$$

$$\text{SIS}(\breve{Y}_{\text{sampled}}, \breve{Y}_{\text{input}}) = \frac{1}{1 + \text{SID}(\breve{Y}_{\text{sampled}}, \breve{Y}_{\text{input}})}, \quad (17)$$

where $y_{\text{sampled}}$ and $y_{\text{input}}$ are vectors representing the spectra of sampled molecules and the input spectra, respectively.

## F RESULTS OF VIBRATIONAL ANALYSIS BY GAUSSIAN16

Figure A5 shows excerpts from the vibrational analysis output of Gaussian16 package (Frisch et al., 2016). The vibrational modes shown on the right correspond to the spectral peaks highlighted on the left. Each table presents displacement vectors (x, y, z) for all atoms in the molecule, indicating how atomic positions change during the corresponding vibrational mode. These modes correspond to spectral peaks that receive high attention from our model, further validating its interpretability.

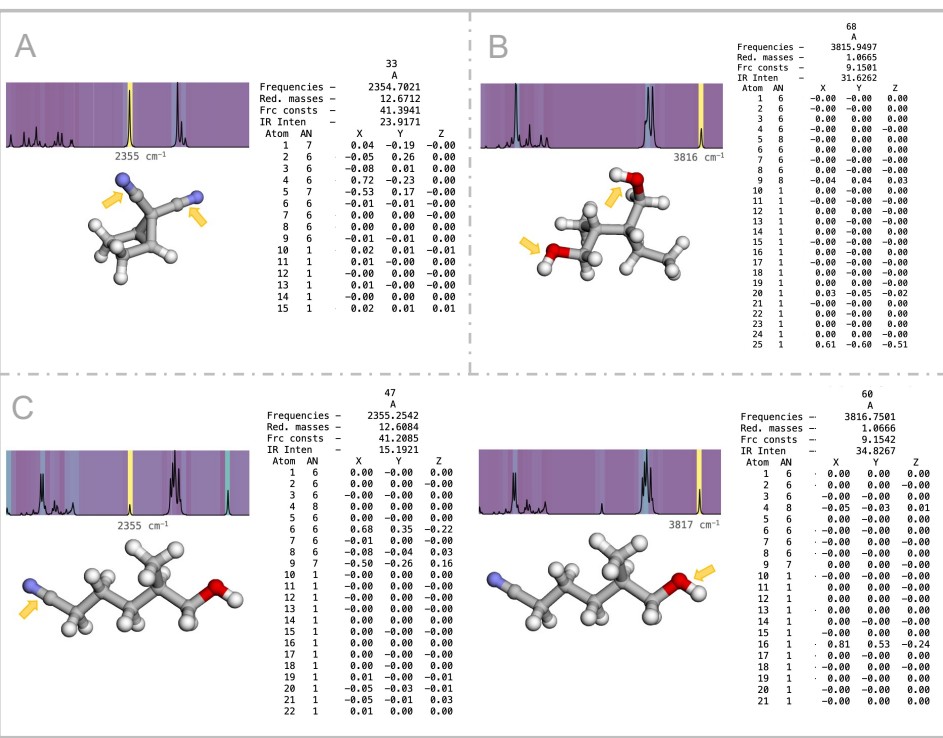

Figure A5: Attention map and quantum mechanical vibrational analysis. **Left**: Cross-attention scores from the model between spectral features and edge features. (For atoms, hydrogen: white, carbon: gray, oxygen: red, nitrogen: blue.) **Right**: Gaussian16 vibrational analysis output. The displacement vectors (x, y, z) indicate atomic movements in each vibrational mode.

## G THE USE OF LARGE LANGUAGE MODELS (LLMS)

We used Large Language Models (LLMs) to aid in polishing the writing of this manuscript. The LLM was employed only for improving the grammar, clarity, and fluency of the text. All scientific ideas, analyses, experiments, and conclusions were conceived and carried out by the authors without assistance from LLMs.

