# OpenReview forum: "Latent Diffusion-based 3D Molecular Recovery from Vibrational Spectra"
_ICLR.cc/2026/Conference — Submitted to ICLR 2026_

### Official Review · Reviewer_sQdo · 2025-10-20

**Soundness:** 3
**Presentation:** 4
**Contribution:** 2
**Rating:** 4
**Confidence:** 3

**Summary:**

The authors aim to reconstruct three-dimensional molecular geometries from infrared (IR) spectra. Unlike prior work that maps spectra to 1D SMILES or 2D graphs, IR-GeoDiff seeks to recover full 3D atomic coordinates. Their method conditions a geometric latent diffusion model on spectral information using a classifier. The model integrates IR features into both node (atomic) and edge (bond) representations through cross-attention with spectral patches and functional group features. The paper evaluates the approach using the Spectral Information Similarity and chemical graph similarity on the QM9S dataset. The authors also find that spectral peaks receiving the highest attention frequently correspond to functional groups.

**Strengths:**

- To the best of my knowledge, this is the first paper that attempts to deduce 3D structure from IR
- Significant performance gains over baseline 3D generative diffusion models (EDM, GEOLDM)

**Weaknesses:**

- Method is essentially a recombination of known techniques
- Assumes that the atom types and the atom count are known beforehand
- Metrics seem problematic

**Questions:**

- How reasonable is it to assume that only a single 3D structure generated an IR spectra? In an experiment there is surely a distribution over structures in a sample, is it not?
- You mention that number of cases where molecules exhibit high graph similarity but low SIS, and vice versa. At the same time, only ~200 test samples were used for spectral metrics due to computational cost of SIS. What would be better metrics?

---

> ### Author Response · Authors · 2025-11-21
> **Rebuttal by Authors (Part 1)**
>
> We thank the reviewer sQdo for the insightful feedback and the recognition that this work represents the first attempt to recover full three dimensional molecular structures directly from IR spectra. Below we respond to the specific concerns raised.
>
> > `w1` **Method is essentially a recombination of known techniques**
>
> We would like to emphasize that the design of our IR-GeoDiff introduces several technical innovations that are specifically motivated by the physics of infrared spectroscopy and go beyond standard combinations of existing components.
>
> First, we use a Transformer-based spectral classifier to extract global spectral features and chemically meaningful functional group features from the input IR spectrum. The functional group classifier is trained jointly to ensure that the spectral representations capture structural information that is known to determine characteristic IR absorption peaks. These spectral and functional group features serve as conditioning signals throughout the denoising process.
>
> Second, rather than simply concatenating conditioning vectors to node features, which is common in prior generation models like EDM and GEOLDM, IR-GeoDiff integrates spectral information through multi-head cross-attention applied to both node and edge representations. The functional group features are injected directly into the edge representations. These functional group features capture local structural motifs that correspond to characteristic IR absorption peaks, and conditioning the edges on them enables the model to emphasize specific bonds and substructures whose vibrational modes dominate the observed spectrum. This physically motivated design allows the diffusion model to fuse spectral and geometric information in a selective and interpretable manner. The ablation studies demonstrate the necessity of these components, and the attention visualization analysis shows that the learned attention aligns with expected vibrational patterns from quantum chemical theory.
>
> Third, we propose a comprehensive evaluation protocol for the spectrum-to-geometry recovery task. Unlike de novo molecular generation or conformer generation, the goal here is to recover geometries that are consistent with a given IR spectrum and to reduce the candidate space as much as possible. Our evaluation metrics assess consistency from both structural and spectral perspectives, and Section 5.3 discusses why both metrics are necessary and complementary.
>
> In addition, we have revised the Related Work section to more clearly articulate the core challenges of spectrum-conditioned 3D structure recovery and to highlight how the design of IR-GeoDiff addresses these challenges.

---

> ### Author Response · Authors · 2025-11-21
> **Rebuttal by Authors (Part 2)**
>
> > `w2`: **Assumes that the atom types and the atom count are known beforehand**
>
> We would like to clarify that assuming the atomic composition to be known is standard in spectroscopy-based structure interpretation and aligns with how IR spectra are used in practice.
>
> In real spectroscopic workflows, IR spectroscopy is primarily applied for qualitative analysis, especially for identifying the presence of specific functional groups (Coates et al., 2000, Clayden et al., 2012), and it provides limited information about the full atomic composition, making it extremely difficult to determine the complete molecular structure based on IR data alone. Therefore, in practice, structure elucidation is rarely attempted using IR spectra in isolation. Specifically, as outlined in [R1] (Chapter 8, Determining the Structure of Organic Molecules from Spectra), the molecular formula is typically established beforehand using other techniques such as elemental analysis or complementary spectroscopic data, and is then used as a known condition when interpreting IR spectra (Line 158-161). Our assumption therefore reflects a realistic, standard, and experimentally grounded setting rather than an artificial constraint.
>
> Moreover, conditioning on the atomic composition is fully consistent with existing work on spectrum-to-structure recovery. Because a single spectrum rarely contains enough information to uniquely determine molecular identity, it is common practice to use the molecular formula as an input condition. This is the case for IR spectrum conditioned structure prediction (Alberts et al., 2024a; Wu et al., 2025; Alberts et al., 2025) (Line 209-210), for mass spectrometry conditioned molecular generation (Bohde et al., 2025) (Line 129-130), and for NMR spectrum inverse problems [R2]. In all of these settings, the atomic composition is treated as part of the available prior information, and the model focuses on resolving the remaining structural ambiguity given that composition.
>
> In summary, **our formulation aligns with how spectra are actually used in laboratory workflows and with how prior spectrum conditioned generative models are designed in the literature.** While our method is not intended to replace all upstream analytical steps required to determine the molecular formula, the assumption of known atomic composition does not severely limit practical usability. Instead, it represents a necessary and standard component of any realistic spectrum-to-structure pipeline.
>
> [R1] Field, L, et al. Organic structures from spectra. John Wiley & Sons, 2013.
>
> [R2] Jonas, Eric. "Deep imitation learning for molecular inverse problems." NeurIPS 2019.
>
> > `w3`: **Metrics seem problematic**
>
> We would appreciate it if the reviewer could specify the issues they found with our evaluation metrics, and which metric(s).
> Then we'd be more than happy to clarify.

---

> > ### Author Response · Authors · 2025-11-21
> > **Rebuttal by Authors (Part 3)**
> >
> > > `Q1` **How reasonable is it to assume that only a single 3D structure generated an IR spectra? In an experiment there is surely a distribution over structures in a sample, is it not?**
> >
> > In common real experimental conditions, an observed IR spectrum is a superposition of contributions from many different conformers, combined through Boltzmann weighting (McGill et al. 2021). In computational chemistry, the simulated experimentental spectrum is typically obtained as a Boltzmann average of over conformer specific spectra[R1, R2]. Every single, distinct conformer in that ensemble theoretically corresponds to its own unique IR spectrum [R3].
> >
> > Our model $p_{\theta}(G|S)$ is designed to address this foundational "one-to-one" theoretical mapping: recovering a single geometry $G$ from its corresponding theoretical spectrum $S$. We argue that solving this task is a necessary and critical first step before tackling the more complex problem of deconvolving experimental spectra that arise from ensembles of conformers.
> >
> > This setting is directly reflected in the QM9S dataset used in our experiments. Each molecule is first optimized to a stable geometry, and its IR spectrum is then computed at that specific geometry, yielding paired data ($G,S$) that instantiate the one to one conformer spectrum relationship.
> >
> > Furthermore, our analysis in Section 5.3 (Figure 4C) empirically confirms that IR spectra are sensitive to conformational changes: even modest geometry changes leading to the formation of intramolecular hydrogen bonds, while leaving the molecular graph unchanged, already lead to noticeable differences in the IR spectrum. This supports the relevance and meaningfulness of studying the conformer level mapping between 3D structure and its associated IR spectrum.
> >
> > [R1] Marton, Gabriel, et al. "An artificial intelligence approach for tackling conformational energy uncertainties in chiroptical spectroscopies." Angewandte Chemie International Edition 62.38 (2023)
> >
> > [R2] Yurenko, Yevgen P., et al. "How many conformers determine the thymidine low-temperature matrix infrared spectrum? DFT and MP2 quantum chemical study." The Journal of Physical Chemistry B 111.32 (2007)
> >
> > [R3] Von Helden, G., et al. "Mid-IR spectra of different conformers of phenylalanine in the gas phase." Physical Chemistry Chemical Physics 10.9 (2008)
> >
> > > `Q2` **You mention that number of cases where molecules exhibit high graph similarity but low SIS, and vice versa. At the same time, only ~200 test samples were used for spectral metrics due to computational cost of SIS. What would be better metrics?**
> >
> > The phenomenon highlighted by the reviewer, where certain molecules achieve high graph similarity but low SIS (and vice versa), demonstrates that structural similarity sim$_g$ and spectral similarity (SIS) are complementary and non-redundant metrics.
> >
> > * sim$_g$ alone is insufficient, because it is insensitive to purely conformational changes (e.g., formation of intramolecular hydrogen bonds) that leave the molecular graph unchanged but cause large shifts in the IR spectrum.
> > * SIS alone is also insufficient, because it may assign similar scores to different carbon scaffolds that share comparable functional-group patterns and vibrational signatures, even though the underlying molecular graphs differ.
> >
> > Therefore, both metrics are important and to our knowledge, this is the best choice. It would be appreciated if the reviwer could point us to any better metric, and we'll be more than happy to engage with discussions and experiments.
> >
> > Regarding the computational cost of SIS, computing spectral similarity requires quantum chemical vibrational analysis for each generated geometry. This is substantially more expensive than computing graph based metrics and is the reason why we report SIS  computed on a subset of 200 test spectra. A natural alternative would be to replace the quantum chemistry (QC) calculation with a fast, deep-learning-based spectral predictor. However, since IR-GeoDiff is, to our knowledge, the first model for this spectrum-to-3D recovery task, we chose to prioritise physical rigour and avoid introducing an additional source of approximation. Using a learned spectral predictor would entangle the generative error of IR-GeoDiff with the prediction error of the surrogate model, making it much harder to interpret whether discrepancies arise from our method or from the metric itself. Therefore, in this initial study we deliberately adopt the QC-based SIS as a “gold-standard” spectral metric (the same level of theory used to construct QM9S), complemented by structural metrics such as sim$_g$ and molecular accuracy. Designing more efficient and principled spectral–structural metrics (e.g., reliable learned surrogates of SIS) is an interesting direction for future work, but is beyond the scope of the present paper.

---

### Official Review · Reviewer_YN2k · 2025-10-28

**Soundness:** 2
**Presentation:** 3
**Contribution:** 2
**Rating:** 2
**Confidence:** 3

**Summary:**

This work appears to be primarily a benchmark study. Building upon previous molecular generation approaches that failed to capture spectral features, the authors aim to address the problem of modeling the distribution of 3D molecular geometries corresponding to a single IR spectrum, under the assumption that atom types and counts are fixed. In the implementation, they construct a corresponding dataset based on QM9 and develop a framework that integrates techniques such as classifier guidance, latent diffusion models (LDM), and GeoLDM to achieve this goal.

**Strengths:**

1. The task is novel and introduces a fresh perspective to the field.
2. The *Background* and *Related Works* sections are well-organized and provide a thorough summary, which is appropriate for a benchmark-focused study.
3. The overall framework maintains good SE(3)-equivariance properties throughout the model design.
4. The integration of spectral features is reasonable and well-supported by the ablation studies.

**Weaknesses:**

1. The paper makes two main claims regarding the task: (1) it aims to model the distribution of 3D molecular geometries corresponding to a single IR spectrum, as stated in the *Abstract* and *Introduction*; and (2) it seeks to learn a probabilistic model $\theta$ that captures the conditional distribution of molecular geometries given an IR spectrum, i.e., $p_{\theta}(G|S)$, as described in the *Preliminaries* section. However, I question the accuracy of this task formulation, since in the actual implementation the authors assume that the atom types $h$ and the atom count $N$ of each molecule are known, and focus solely on modeling the conditional distribution over atomic coordinates $x$. Therefore, the model effectively learns $p_{\theta}(x|S,h)$ rather than $p_{\theta}(G|S)$, and the paper should have explicitly clarified at the outset what the true modeling objective of the task is.
2. The benchmark dataset is undersized: it relies solely on QM9, which includes molecules with at most nine heavy atoms. Contemporary 3D molecular tasks typically involve much larger molecules—e.g., those in GEOM-Drugs or PCQM4Mv2 average tens to hundreds of heavy atoms—so larger, multiscale benchmarks would be more representative and comprehensive.
3. The overall framework is primarily built upon several well-established methods, including classifier guidance, latent diffusion models (LDM), and GEOLDM. Although this design is adequate for a benchmark-oriented study, it offers limited novelty beyond existing approaches.
4. The baselines used in this work are relatively outdated and limited in number. In the field of molecular generation, many newer and more powerful methods beyond EDM and GeoLDM have been proposed, which could be adapted for comparison. At the very least, some of these recent approaches should have been considered to provide a more comprehensive evaluation.
5. The results and implementation of EDM and GeoLDM appear questionable and inconsistent. In your setting, both the atom types and atom counts are provided as part of the input, whereas for EDM and GeoLDM, only the number of atoms is fixed and the atom types are omitted—a significant loss of information. This discrepancy raises concerns about the fairness of the comparison. Please refer to **Question 2** for further discussion.
6. There are no formulations provided for any of the evaluation metrics in the manuscript.

**Questions:**

1. The manuscript states that *“in the subsequent diffusion training stage, the spectral classifier is frozen to ensure a stable and consistent conditioning signal, while the autoencoder remains learnable.”* Why does the autoencoder remain learnable during the diffusion training stage? Is there any ablation study or experimental evidence supporting this design choice?
2. The manuscript claims that *“denoising networks jointly model position- and feature-level noise, which precludes specifying atom types as inputs for EDM and GeoLDM.”* Why is this the case? The paper provides neither a theoretical formulation nor an architectural illustration to justify this restriction.
3. In the definition of *molecular accuracy*, it is stated that *“if at least one sampled molecule exactly matches the reference structure.”* How is “matches” defined in this context? Does it refer to exact coordinate alignment, atom-type correspondence, or another structural similarity measure?
4. During evaluation, is there any metric that measures spatial deviation—such as the root mean square deviation (RMSD)—between the generated positions $ x^{\prime}$ and the reference positions $ x $?

---

> ### Author Response · Authors · 2025-11-21
> **Rebuttal by Authors (Part 1)**
>
> >`Summary` This work appears to be primarily a benchmark study. Building upon previous molecular generation approaches that failed to capture spectral features, the authors aim to address the problem of modeling the distribution of 3D molecular geometries corresponding to a single IR spectrum, under the assumption that atom types and counts are fixed. In the implementation, they **construct a corresponding dataset based on QM9** and develop a framework that integrates techniques such as **classifier guidance**, latent diffusion models (LDM), and GeoLDM to achieve this goal.
>
> We thank Reviewer YN2k for their time and effort in reviewing our work. We appreciate the recognition of the novelty of the proposed task, the clarity of the background and related work sections, the SE(3)-equivariance of the framework, and the effectiveness of the spectral feature integration. We would like to clarify **two factual misunderstandings** mentioned in the summary.
>
> 1. Regarding the dataset, we did not construct a new dataset for this study. We use the existing QM9S dataset developed by Zou et al. (2023), as detailed in Line 343-348.
> 2. Our method does not employ classifier guidance. Classifier guidance typically refers to using the gradients of a classifier at sampling time to steer the denoising process. In contrast, as described in Section 4.1, the spectral classifier $\tau_\theta$ is used only to extract static conditioning vectors, namely the spectral features $S$ and functional group features $M$. These embeddings are passed as fixed inputs to the denoising network $\epsilon_{\theta}$, and we do not use any classifier gradients during sampling.
>
> Below we respond to the specific concerns raised.
>
> > `w1` **The paper makes two main claims regarding the task: (1) it aims to model the distribution of 3D molecular geometries corresponding to a single IR spectrum, as stated in the Abstract and Introduction; and (2) it seeks to learn a probabilistic model that captures the conditional distribution of molecular geometries given an IR spectrum, i.e., $p_\theta(G|S)$ , as described in the Preliminaries section. However, I question the accuracy of this task formulation, since in the actual implementation the authors assume that the atom types and the atom count of each molecule are known, and focus solely on modeling the conditional distribution over atomic coordinates. Therefore, the model effectively learns $p_\theta(x|S, h)$ rather than $p_\theta(G|S)$, and the paper should have explicitly clarified at the outset what the true modeling objective of the task is.**
>
> We thank the reviewer for pointing out this notational detail. We would like to clarify that our original manuscript already stated this assumption in Section 3.1: "*Therefore, in our setting, we assume that the atom types h and the atom count N of a molecule are known, and focus on modelling the conditional distribution over atomic coordinates x*"(Line161-163). However, we agree that updating the mathematical notation to explicitly reflect this assumption could improve precision. We have added the following clarification, highlighted in blue in the revised version: "*In other words, our problem can be formulated as $p_\theta(\mathbf{x}|S,\mathbf{h})$.*"

---

> ### Author Response · Authors · 2025-11-21
> **Rebuttal by Authors (Part 2)**
>
> > `w2`: **The benchmark dataset is undersized: it relies solely on QM9, which includes molecules with at most nine heavy atoms. Contemporary 3D molecular tasks typically involve much larger molecules—e.g., those in GEOM-Drugs or PCQM4Mv2 average tens to hundreds of heavy atoms—so larger, multiscale benchmarks would be more representative and comprehensive.**
>
> First, we would like to clarify that the benchmark used in this work is not constructed by us. We use the publicly available QM9S dataset developed by Zou et al. (2023), which provides IR spectra together with corresponding 3D molecular geometries (Lines 343–348).
>
> Although datasets such as GEOM-Drugs and PCQM4Mv2 contain much larger and more diverse molecules and are widely used in contemporary 3D molecular benchmarks, these datasets **do not** include IR spectra, which are essential for our spectrum-to-geometry recovery task. Conversely, datasets used in prior IR-to-structure studies typically provide IR spectra but lack 3D geometries. At the time of submission, QM9S was the only available dataset we found that simultaneously offers IR spectra and corresponding 3D molecular structures.
>
> During the rebuttal period, we discovered a recently published dataset, QMe14S [R1], which contains 186,102 molecules with IR spectra and corresponding 3D geometries. QMe14S is constructed by combining the QM9S molecules with additional 56,285 molecules selected from PubChem, resulting in a dataset that spans 14 elements and covers substantially larger and more chemically diverse structures than QM9S.
>
> **As suggested, we further perform experiments on this new dataset.** Specifically, we removed all entries overlapping with the QM9S training split to avoid any data leakage. We then applied additional quality control filtering to remove molecules with problematic geometries. The resulting subset contains 53,344 molecules. We shuffled this subset, selected 1000 molecules as the test set, and split the remaining data into training and validation sets at a 95:5 ratio.
>
> We first defined 26 common functional groups and trained a spectral classifier on the combined QM9S and QMe14S training sets for 100 epochs. We then finetuned the autoencoder and diffusion model, starting from QM9S pretrained weights, on the QMe14S training and validation set. The autoencoder was trained for 150 epochs and the diffusion model was trained for 100 epochs. For evaluation, we sampled 50 molecules for each spectrum in the 1000 molecule QMe14S test set. The results are as follows:
>
> | Data set | sim$_g$ | max sim$_g$ | mol acc (%)|
> | -------- | -------- | -------- | -------- |
> | QM9S     | 0.672 | 0.969 | 90.6 |
> | QMe14S subset | 0.569 | 0.918 | 84.3 |
>
> Although QMe14S contains molecules with larger size and greater structural diversity, the performance of IR-GeoDiff decreases only moderately. Importantly, IR-GeoDiff on QMe14S still performs markedly above the EDM and GeoLDM baselines evaluated on the simpler QM9S dataset. This demonstrates that IR-GeoDiff generalizes meaningfully beyond the QM9S domain and highlights the potential of IR-guided diffusion models to scale beyond small-molecule benchmarks.
>
> [R1] Yuan M, Zou Z, Luo Y, Jiang J, Hu W. QMe14S: A Comprehensive and Efficient Spectral Data Set for Small Organic Molecules. The Journal of Physical Chemistry Letters, 16(16):3972-9, 2025.

---

> > ### Author Response · Authors · 2025-11-21
> > **Rebuttal by Authors (Part 3)**
> >
> > > `w3` **The overall framework is primarily built upon several well-established methods, including classifier guidance, latent diffusion models (LDM), and GEOLDM. Although this design is adequate for a benchmark-oriented study, it offers limited novelty beyond existing approaches.**
> >
> > We would like to emphasize that the design of our IR-GeoDiff introduces several technical innovations that are specifically motivated by the physics of infrared spectroscopy and go beyond standard combinations of existing components.
> >
> > First, we use a Transformer-based spectral classifier to extract global spectral features and chemically meaningful functional group features from the input IR spectrum. The functional group classifier is trained jointly to ensure that the spectral representations capture structural information that is known to determine characteristic IR absorption peaks. These spectral and functional group features serve as conditioning signals throughout the denoising process.
> >
> > Second, rather than simply concatenating conditioning vectors to node features, which is common in prior generation models like EDM and GEOLDM, IR-GeoDiff integrates spectral information through multi-head cross-attention applied to both node and edge representations. The functional group features are injected directly into the edge representations. These functional group features capture local structural motifs that correspond to characteristic IR absorption peaks, and conditioning the edges on them enables the model to emphasize specific bonds and substructures whose vibrational modes dominate the observed spectrum. This physically motivated design allows the diffusion model to fuse spectral and geometric information in a selective and interpretable manner. The ablation studies demonstrate the necessity of these components, and the attention visualization analysis shows that the learned attention aligns with expected vibrational patterns from quantum chemical theory.
> >
> > Third, we propose a comprehensive evaluation protocol for the spectrum-to-geometry recovery task. Unlike de novo molecular generation or conformer generation, the goal here is to recover geometries that are consistent with a given IR spectrum and to reduce the candidate space as much as possible. Our evaluation metrics assess consistency from both structural and spectral perspectives, and Section 5.3 discusses why both metrics are necessary and complementary.
> >
> > In addition, we have revised the Related Work section to more clearly articulate the core challenges of spectrum conditioned 3D structure recovery and to highlight how the design of IR-GeoDiff addresses these challenges.

---

> ### Author Response · Authors · 2025-11-21
> **Rebuttal by Authors (Part 4)**
>
> > `w4` **The baselines used in this work are relatively outdated and limited in number. In the field of molecular generation, many newer and more powerful methods beyond EDM and GEOLDM have been proposed, which could be adapted for comparison. At the very least, some of these recent approaches should have been considered to provide a more comprehensive evaluation.**
>
> We chose EDM and GEOLDM as baselines because these two methods are widely regarded as the foundational architectures for 3D molecular diffusion and represent the two major design paradigms in the field: diffusion directly in coordinate space (EDM) and diffusion in a learned latent space (GEOLDM). Both have become standard baselines in subsequent 3D molecular diffusion papers, making them a natural starting point for evaluating our spectrum-conditioned recovery task.
>
> We note that EDM and GEOLDM were originally proposed for unconditional or property-conditioned de novo generation. As discussed in the related work section, recent models in 3D molecular diffusion also consider tasks such as graph-to-3D conformer prediction and conditioning on complex geometric modalities including protein pockets or reference 3D structures. None of these methods are designed for recovering a 3D molecular geometry from a single IR spectrum. At submission time, we therefore prioritised adapting EDM and GEOLDM, since they are well established baselines in the field and their conditioning mechanism based on feature concatenation can be extended to IR spectral features with minimal architectural modification.
>
> We thank the reviewer’s suggestion to include more recent architectures. As suggested, we therefore further compare IR-GeoDiff to two recent equivariant diffusion models:
> * **GFMDiff**: Geometric-Facilitated Denoising Diffusion Model for 3D Molecule Generation (AAAI 2024). GFMDiff incorporates both pairwise distances and triplet-wise angles in the denoising kernel to capture higher-order local geometric interactions. These multi-body patterns are closely related to functional-group-specific vibrational modes, making GFMDiff a relevant architecture to test in our IR-conditioned setting.
> * **END**: Equivariant Neural Diffusion for Molecule Generation (NeurIPS 2024). END is an Euclidean-equivariant diffusion model with a learnable forward process. END evaluates composition-conditioned and substructure-conditioned generation, which require tighter control over both global composition and local motifs and are thus conceptually closer to our spectrum-guided recovery task than simple property-based conditioning.
>
> At this rebuttal stage, we have trained both models for 200 epochs, and the preliminary results are shown in the table below, where we also report the result of our method trained for 200 epochs. All models are evaluated by using 30 generated samples per test case (the computationally expensive SIS metrics were not computed here). These results already indicate that recent diffusion models do not trivially transfer to IR-spectrum-to-3D recovery, reinforcing the need for architectures explicitly designed for spectrum-conditioned structure reconstruction. The full results for both models will be reported in the final version.
>
> | Model (trained 200 epochs) | sim$_g$ | max sim$_g$ | Mol acc(%) |
> | -------- | -------- | -------- | -------- |
> | GFMDiff | 0.280 | 0.543 | 10.0 |
> | END | 0.127 | 0.325 | 0.5  |
> | Ours | **0.515** | **0.876** | **67.5** |
>
> > `w6` **There are no formulations provided for any of the evaluation metrics in the manuscript.**
>
> We appreciate the reviewer’s comment. Our evaluation protocol combines structural and spectral consistency and is tailored to the spectrum-to-geometry recovery task. The individual similarity measures used within this protocol are based on standard formulations from prior works (McGill et al. 2021;  Adams & Coley 2023; Chen et al. 2023; (Alberts et al., 2024a; Wu et al., 2025; Alberts et al., 2025), as detailed in Section 4.5.
>
> For structural consistency, graph similarity sim$_g$ is defined as the Tanimoto similarity between the Morgan fingerprints of the sampled molecule and the ground truth molecule (Line 320-321).
>
> For spectral consistency, Spectral Information Similarity (SIS) is computed based on the Spectral Information Divergence (SID) introduced by McGill et al.（2021). Given two normalized spectra, the spectrum of sampled molecule $Y_{s}$ and the input spectrum $Y_{in}$ , SID and SIS are defined as:
> $$
> SID(Y_s, Y_{in}) = \sum_i y_{s,i} \ln \dfrac{y_{s,i}}{y_{{in},i}} + y_{in, i}\ln \dfrac{y_{in,i}}{y_{s,i}},
> SIS(Y_s, Y_{in})=\dfrac{1}{1+SID(Y_s, Y_{in})}
> $$
>
>
> We have now included this full formulation in Appendix E.5, highlighted in blue.

---

> ### Author Response · Authors · 2025-11-21
> **Rebuttal by Authors (Part 5)**
>
> > `Q1`: **The manuscript states that “in the subsequent diffusion training stage, the spectral classifier is frozen to ensure a stable and consistent conditioning signal, while the autoencoder remains learnable.” Why does the autoencoder remain learnable during the diffusion training stage? Is there any ablation study or experimental evidence supporting this design choice?**
>
> We implemented a two-stage pipeline: (1) Pretraining the autoencoder, and (2) Jointly optimizing the autoencoder and diffusion model.
>
> From a theoretical perspective, our design is based on the analysis of latent diffusion models in GEOLDM. Their Theorem 4.2 shows that optimizing the combined objective ($L_{AE} + L_{LDM}$) constitutes maximizing a SE(3)-invariant variational lower bound (VLB) on the data log-likelihood. In our setting, this suggests that there is no theoretical obstacle to continuing to update the autoencoder together with the latent diffusion model, since both components contribute jointly to the same likelihood bound. Allowing the autoencoder to remain learnable lets the latent space adapt to the spectrum conditioned diffusion process, instead of forcing the diffusion model to fit a fixed latent distribution that was optimized without the IR conditioning.
>
> Empirically, we have observed that this joint optimization is stable and that the reconstruction quality of the autoencoder improves after the diffusion training stage. The table below reports the mean squared error (MSE) between the input 3D coordinates and the reconstructed geometries on the QM9S test set, both after autoencoder pretraining and after the full diffusion training:
>
> |  | After AE training (Å$^{2}$) | After DM training (Å$^{2}$) |
> | -------- | -------- | -------- |
> | MSE (test set)  | $2.98\times10^{-3}$     |$4.79\times10^{-6}$|
>
> These results indicate that keeping the autoencoder learnable during diffusion training does not degrade the latent representation. Instead, it further refines the latent space so that it remains compatible with both faithful reconstruction and the spectrum conditioned generative objective.
> > `w5`: **The results and implementation of EDM and GeoLDM appear questionable and inconsistent... This discrepancy raises concerns about the fairness of the comparison...**
> >
> > `Q2` **The manuscript claims that “denoising networks jointly ...” Why is this the case? The paper provides neither a theoretical formulation nor an architectural illustration to justify this restriction.**
>
> We appreciate the reviewer’s request for clarification, which touches on a **fundamental difference between our recovery task and the de novo generation setting addressed by EDM and GEOLDM**.
>
> In de novo generation models such as EDM and GeoLDM, both atomic coordinates $\mathbf{x}$ and atomic features $\mathbf{h}$ (i.e., atom types) are treated as variables to be generated. The model is trained to learn the joint distribution $p_\theta(\mathbf{z_x}, \mathbf{z_h})$. Their denoise network $\epsilon_{\theta}$ predict noise $\epsilon=[\epsilon_x, \epsilon_h]$ for both the position and feature components simultaneously. During the forward diffsuion process, noise is added to both $\mathbf{z_x}$ and $\mathbf{z_h}$, and sampling starts from a pair of fully noisy latents ($\mathbf{z_x}^T, \mathbf{z_h}^T$) from a standard Gaussian prior.
>
> Consequently, the denoising network $\epsilon_{\theta}$ is trained under the assumption that both inputs are noisy. It expects a noisy feature latent $\mathbf{z_h}^t$ at every diffusion step and learns to remove noise from both $\mathbf{z_x}^t$ and $\mathbf{z_h}^t$ jointly. During sampling, simply “fixing” the atom types by feeding a clean $\mathbf{z_h}$ while only denoising $\mathbf{z_x}$ would put the model in an out of distribution regime that the denoising network has never seen during training, and the current architecture is not designed to interpret $\mathbf{z_h}$ as a non-noised conditioning variable. A principled way to incorporate known atom types into EDM or GEOLDM would require redefining the forward process and objective so that the model explicitly learns a conditional distribution $p_\theta(\mathbf{z_x} | \mathbf{z_h})$ rather than the original joint distribution $p_\theta(\mathbf{z_x}, \mathbf{z_h})$. This is a non trivial modification of the original methods and goes beyond a simple change of inputs at sampling time.
>
> By contrast, in IR-GeoDiff, we explicitly adopt the spectrum conditioned recovery setting where $\mathbf{h}$ and $N$ are assumed to be known, as defined in Section 3.1. Our architecture is therefore designed from the outset for this conditional scenario:
>
> 1.   Diffusion is applied only to the positions latent $\mathbf{z_x}$.
> 2.  The known atom types $\mathbf{z_h}$ are not diffused. Instead, they are fed as a fixed condition into the denoising network at every step along with the spectral features $S$ and functional group features $M$, i.e. $\epsilon_{\theta}(\mathbf{z_x}^t, t, S, M, \mathbf{z_h})$

---

> ### Author Response · Authors · 2025-11-21
> **Rebuttal by Authors (Part 6)**
>
> > `Q3` **In the definition of molecular accuracy, it is stated that “if at least one sampled molecule exactly matches the reference structure.” How is “matches” defined in this context? Does it refer to exact coordinate alignment, atom-type correspondence, or another structural similarity measure?**
>
> Following previous IR to structure work (Alberts et al., 2024a; Wu et al., 2025; Alberts et al., 2025), we define a molecule as correct *if its canonical SMILES string exactly matches the canonical SMILES of the ground truth reference molecule*. Canonical SMILES provides a unique representation of the molecular graph, which allows us to determine chemical identity in a consistent manner across all sampled molecules. This definition focuses on graph level correctness rather than coordinate alignment, because molecular accuracy measures whether the correct bonding pattern has been recovered.
>
> > `Q4`. **During evaluation, is there any metric that measures spatial deviation—such as the root mean square deviation (RMSD)—between the generated positions and the reference positions?**
>
> We did not use RMSD as a primary evaluation metric, and this is a deliberate choice rather than an omission. RMSD is a common metric for tasks like conformer generation, where the molecular graph is given and the goal is to reproduce a particular low-energy conformer. However, in our setting, the input is only the IR spectrum (along with the given atomic composition), and the model must recover both the molecular graph and the 3D geometry. This leads to two qualitatively different cases:
>
> 1. **The recovered molecule has a different graph from the reference.** In this case, RMSD is not very informative: two molecules with different connectivity cannot be regarded as the “same conformer”, so a large RMSD is expected and does not provide a meaningful notion of partial success. Instead, molecular accuracy and graph similarity are more meaningful, since they directly assess whether the recovered chemical structure matches the reference. These metrics capture chemically meaningful differences (e.g. whether key local patterns such as functional groups are correctly recovered), which RMSD cannot reveal.
> 2. **The recovered molecule has the same graph as the reference.** Here, RMSD can be small even when the conformation is not consistent with the IR spectrum. For the case in Figure 4C (bottom), the recovered geometry have a low RMSD of 0.45 Å relative to the reference, below the commonly used 0.5 Å conformer matching threshold for QM9 dataset [R1, R2]. However, the spectral information similarity (SIS) is only 0.380, indicating a significant spectral mismatch. The discrepancy is mainly caused by small conformational changes that form intramolecular hydrogen bonds, which are known to induce notable shifts in IR spectra. In such cases, RMSD would misleadingly suggest success, whereas SIS correctly identifies the spectral inconsistency.
>
> In contrast, SIS is directly aligned with the objective of spectrum-to-geometry recovery. IR spectra are computed from the vibrational modes of a molecule using the 3D coordinates as input. Any geometrical or conformational deviation that affects the vibrational frequencies will be reflected in the spectrum and therefore penalized by SIS. For this reason, SIS is a more task relevant metric for evaluating geometric correctness in our setting. As discussed in Section 5.3 (Limitations), SIS and our structural metrics (graph similarity and molecular accuracy) together evaluate whether the recovered structures are both chemically correct and spectrally consistent.
>
> [R1] Ganea, Octavian, et al. "Geomol: Torsional geometric generation of molecular 3d conformer ensembles." NeurIPS 2021
>
> [R2] Xu, Minkai, et al. "GeoDiff: A Geometric Diffusion Model for Molecular Conformation Generation." ICLR 2022

---

> ### Comment · Reviewer_YN2k · 2025-11-22
> **Response for Authors' Updates**
>
> Thank you for the updates and for your hard work.
>
> First, I would like to apologize for the previous misunderstandings regarding the **dataset construction** and **classifier** issues.
>
> After carefully reading your revisions, some of my earlier concerns have been partially addressed. However, I still have the following concerns:
>
> 1. As you also acknowledged, the task is to model $p_{\theta}(x \mid S, h)$. Therefore, I do not fully understand why you choose **de novo molecular generation** baselines (**EDM** and **GeoLDM**). Under your setting, the atom types and counts are fixed, which makes it inherently unfair to these baselines, since you only fix the atom counts for these two methods. For the newly added **GFMDiff** and **EnD**, are they treated in exactly the same way?
>
>    In your **Rebuttal by Authors (Part 5)**, I am fully aware of the modeling process of **EDM** and **GeoLDM**, i.e., learning the joint distribution $p_{\theta}(Z_x, Z_h)$ (denoising both). However, this alone is not a sufficient reason to directly compare your method with them in this setting, because **the loss of atom-type information is substantial**. Either (i) you redefine the forward process and training objective of these baselines and retrain them specifically for your task, or (ii) you select baselines that are actually designed for **molecular conformation generation**, such as **GeoDiff**, **ET-Flow** (with radius-cutoff bonds), etc., which would be more appropriate and convincing.
>
> 2. Are your spectral features and chemically meaningful functional group features also used as conditions in **EDM** and **GeoLDM**? If so, could you clarify how they are incorporated? If not, this may further exacerbate the unfairness of the comparison.
>
> 3. Regarding my Question 1, thank you again for your response. I am not asking for further discussion on this point. However, there is an [interesting related issue](https://github.com/MinkaiXu/GeoLDM/issues/6) that might be helpful for your future investigation.
>
> 4. How are atomic connections (bonds) determined after obtaining the denoised coordinates $x$? I am wondering if I missed this part in the paper or supplementary materials.

---

> > ### Author Response · Authors · 2025-11-23
> > **Response to Follow-up Questions (Part 1)**
> >
> > We sincerely appreciate reviewer YN2K for their prompt and detailed feedback. Below we respond to the specific concerns raised.
> > > `c1` As you also acknowledged, the task is to model $p(c|S,h)$... For the newly added GFMDiff and EnD, are they treated in exactly the same way? ... Either (i) you redefine the forward process and training objective of these baselines and retrain them specifically for your task, or (ii) you select baselines that are actually designed for molecular conformation generation, such as GeoDiff, ET-Flow (with radius-cutoff bonds), etc., which would be more appropriate and convincing.
> >
> > We first clarify that the newly added GFMDiff and EnD are treated in exactly the same way as EDM and GeoLDM where the atom counts are fixed and atom types are unconstrained.
> >
> > Regarding the reviewer’s suggestion, we respectfully clarify why the two proposed options are not appropriate baselines for our task.
> > 1. **Redefine the forward process and training objective of EDM/GEOLDM**:
> > Redefining the forward diffusion process and training objective so that these models can condition on fixed atom types would fundamentally alter their generative formulation. Such modifications would no longer constitute evaluating the baselines as what they are, but rather constructing new models inspired by EDM/GEOLDM. This would undermine the purpose of using baselines, which is to assess whether existing 3D diffusion frameworks when applied in their canonical formulation, are sufficient for the spectrum-to-structure recovery task, or whether new architectures and diffusion parameterizations are needed. Our results indicate that the canonical baselines struggle in this setting, motivating the development of specialized models such as IR-GeoDiff.
> > 2. **Compare to baselines designed for molecular conformation generation**:
> > For the models designed for the conformation generation task, the input is the molecular graph $G$, and the goal is to model $p_\theta(C|G)$. The conditioning graph not only includes the information of atomic composition but also supplies **explicit atom connectivity**. However, in our setting, the molecular graph is unknown and must be inferred from the IR spectrum. The spectrum contains only implicit information about connectivity through vibrational signatures. Substituting the graph $G$ with the IR spectrum $S$ is therefore **infeasible**: the conditioning variables are fundamentally different in nature and information content. Using a conformation-generation model as a baseline would implicitly assume that the connectivity is already known, which bypasses the central difficulty of our task. This is also why the graph-based metric sim$_g$ is essential for our evaluation
> >
> > > `c2` Are your spectral features and chemically meaningful functional group features also used as conditions in EDM and GeoLDM? If so, could you clarify how they are incorporated? If not, this may further exacerbate the unfairness of the comparison.
> >
> > As mentioned in Line 363-364 and detailed in Appendix B.3, we follow the conditioning mechanisms used in EDM and GeoLDM, namely concatenating the spectral feature vector to the node representations for both methods.
> >
> > We note that functional group (FG) features are not inluded  in EDM and GeoLDM, but we respectfully disagree that this creates an unfair comparison, for the following reasons:
> > 1. The FG features used in IR-GeoDiff are not additional ground-truth annotations but the predictions derived solely from the input IR spectrum. Thus, IR-GeoDiff does not have access to information unavailable to EDM or GeoLDM.
> > 2. FG features in IR-GeoDiff are incorporated via cross-attention into edge representations. By contrast, EDM and GeoLDM rely on node-level feature concatenation as their conditioning mechanism, and lack an explicit mechanism to inject specific edge-level conditions. For fairness, we adopt the conditioning mechanisms that each baseline natively supports.
> > 3. Our ablation study (Table 1) shows that IR-GeoDiff without FG conditioning (“w/o fg ca”) still surpasses EDM and GeoLDM across all spectral and structural metrics. This confirms that the performance gain does not originate from FG features themselves, but from the overall architectural design that better aligns spectral information with 3D geometry.

---

> > ### Author Response · Authors · 2025-11-23
> > **Response to Follow-up Questions (Part 2)**
> >
> > > `c3` Regarding my Question 1, thank you again for your response. I am not asking for further discussion on this point. However, there is an interesting related issue that might be helpful for your future investigation.
> >
> > We sincerely appreciate the reviewer’s recognition of our response and for sharing this insightful discussion regarding the potential identity-mapping behavior in GeoLDM’s encoder. Observations about how latent diffusion models may preserve explicit coordinate information in their latent space provide a valuable perspective for understanding the behaviour of geometric latent diffusion models. We will keep this in mind as a helpful reference when analysing and further refining the topology of our spectrum-conditioned latent space in future work.
> >
> > > `c4` How are atomic connections (bonds) determined after obtaining the denoised coordinates $x$? I am wondering if I missed this part in the paper or supplementary materials.
> >
> > We appreciate the reviewer’s question. The procedure for determining atomic connections from the denoised coordinates was indeed provided in Appendix D.1, and we briefly summarise it here for clarity:
> >
> > After obtaining the denoised 3D coordinates, we reconstruct the molecular graph using a valence-guided bond assignment algorithm. We first construct an initial adjacency matrix by connecting atom pairs whose distances fall below element-specific single-bond thresholds. We then iteratively update bond orders to satisfy atomic valence constraints: terminal atoms are saturated first, aromatic rings are identified using Hückel’s rule, and remaining atoms have their bond orders adjusted based on unsaturation and distance criteria. If inconsistencies remain, the most deviant bond is removed and the procedure is repeated until all atoms satisfy their valence or no further updates are possible.

---

> > > ### Comment · Reviewer_YN2k · 2025-11-24
> > > **New Responses**
> > >
> > > Thank you for the updated responses.
> > >
> > > 1. **Regarding Point 1:** I fully understand your statement that *“Such modifications would no longer constitute evaluating the baselines as what they are, and would undermine the purpose of using baselines.”* However, this still does not fully convince me.
> > >
> > >    My point is that you can keep the original EDM/GeoLDM unchanged **but additionally introduce variant versions** (i.e., new models inspired by EDM/GeoLDM) to specifically address the issue that *“the atom counts are fixed and atom types are unconstrained.”*  If these variants perform significantly worse than your proposed method, that already strengthens your contribution. Otherwise, as I mentioned earlier, certain molecular conformation baselines can be adapted—for example, by replacing chemical bonds with radius-cutoff bonds to avoid  introducing **explicit atom connectivity**.
> > >
> > > 2. **Regarding Point 2:**  Your explanation is generally reasonable and mostly clear.
> > >
> > > 3. **Regarding Point 4:**  I apologize for overlooking parts of your Appendix. However, I believe that the “Our Approach” section does not contain reference ro Appendix D or discussion about how the **final molecule** is obtained. This step is essential. From what I can see, the description seems to end at sampling a geometry $G = <x, h>$, without explaining how a complete molecule is reconstructed.

---

> > > > ### Author Response · Authors · 2025-11-26
> > > > **Response to New Follow-up Comments**
> > > >
> > > > We sincerely appreciate reviewer YN2K for their prompt and detailed feedback. Below we respond to the specific concerns raised.
> > > > > **Regarding Point 1**: I fully understand your statement that “Such modifications would no longer constitute evaluating the baselines as what they are, and would undermine the purpose of using baselines.” However, this still does not fully convince me.
> > > > > My point is that you can keep the original EDM/GeoLDM unchanged but additionally introduce variant versions (i.e., new models inspired by EDM/GeoLDM) to specifically address the issue that “the atom counts are fixed and atom types are unconstrained.” If these variants perform significantly worse than your proposed method, that already strengthens your contribution. Otherwise, as I mentioned earlier, certain molecular conformation baselines can be adapted—for example, by replacing chemical bonds with radius-cutoff bonds to avoid introducing explicit atom connectivity.
> > > >
> > > > We thank the reviewer for further detailed explanation. We now better understand the reviewer’s suggestion of introducing variant versions of EDM and GEOLDM that constrain atom types while keeping the original models unchanged. We agree that such variants can offer an informative perspective on the role of atomic-type constraints.
> > > >
> > > > As discussed before, sampling in GEOLDM starts from a pair of fully noisy latents $(\mathbf{z_x}^T, \mathbf{z_h}^T)$ drawn from a standard Gaussian prior. If atom types are constrained, one would instead require noisy $\mathbf{z_x}^T$ but clean $\mathbf{z_h}$, i.e. $(\mathbf{z_x}^T, \mathbf{z_h})$. However, the autoencoder in GEOLDM encodes a clean pair $(\mathbf{x}, \mathbf{h})$ into $(\mathbf{z_x}, \mathbf{z_h})$. Since $\mathbf{x}$ is unknown during sampling, the clean $\mathbf{z_h}$ cannot be obtained. Thus, a new autoencoder is required to obtain $\mathbf{z_h}$ independent of $\mathbf{x}$. To ensure the fairness of comparison as much as possible, we therefore adopt the exact same latent initialization scheme of IR-GeoDiff as detailed in Section 4.2.
> > > >
> > > > These experiments for both variants of EDM and GEOLDM are currently running, and we will update the results as soon as they are completed.
> > > >
> > > > > **Regarding Point 2**: Your explanation is generally reasonable and mostly clear.
> > > >
> > > > We thank the reviewer for the recognition of our response, and glad that this point has been addressed.
> > > >
> > > > > **Regarding Point 4**: I apologize for overlooking parts of your Appendix. However, I believe that the “Our Approach” section does not contain reference ro Appendix D or discussion about how the final molecule is obtained. This step is essential. From what I can see, the description seems to end at sampling a geometry, without explaining how a complete molecule is reconstructed.
> > > >
> > > > We thank the reviewer for pointing this out, and we apologize for the lack of clarity in the main text. We agree that explicitly stating how the sampled geometries are converted into complete molecular structures is essential.
> > > >
> > > > We have revised the manuscript by adding the following sentence in Section 4.5 (highlighted in blue), "*After sampling geometries $G=<\mathbf{x}, \mathbf{h}>$, we further convert them into molecular graphs using a valence-guided bond assignment algorithm, as detailed in Appendix D.1.*"

---

> > > > ### Author Response · Authors · 2025-11-27
> > > > **Response to New Follow-up Comments (Results Update)**
> > > >
> > > > We would like to thank the reviewer again for their valuable suggestion and patience. **Following the reviewer’s guidance, we have trained both the EDM-inspired and the GeoLDM-inspired variants for 200 epochs, and we present the preliminary results in the table below**. For a fair comparison, we also report the performance of our method trained for the same number of epochs. All models are evaluated using 30 samples per test case (the computationally expensive SIS metrics are omitted at this stage). These preliminary results indicate that the physics-motivated design of our method leads to stronger performance. We will include the full experimental results in the final version of the manuscript to further strengthen our contribution.
> > > >
> > > > | Model (trained 200 epochs) | sim$_g$ | max sim$_g$ | Mol acc(%) |
> > > > | -------- | -------- | -------- | -------- |
> > > > | EDM variant | 0.335 |  0.680 | 25.5 |
> > > > | GEOLDM variant | 0.412 | 0.794 | 47.5 |
> > > > | Ours | **0.515** | **0.876** | **67.5** |

---

### Official Review · Reviewer_x6jw · 2025-10-31

**Soundness:** 2
**Presentation:** 3
**Contribution:** 3
**Rating:** 4
**Confidence:** 4

**Summary:**

This paper proposes IR-GeoDiff, a latent diffusion model designed to reconstruct three-dimensional molecular geometries directly from one-dimensional infrared spectra. Existing methods, which typically generate one-dimensional SMILES strings or two-dimensional molecular graphs, fail to capture the intrinsic relationship between infrared spectra and molecular geometry. IR-GeoDiff addresses this limitation through two key innovations: (1) It employs a multi-head cross-attention mechanism to integrate infrared spectral features into the node (atom) and edge (bond) features of the molecular geometry; (2) It utilizes a Transformer-based spectral classifier to simultaneously extract spectral and functional group features, which serve as conditional inputs for the generation process.
The model was evaluated based on both spectral and structural similarity metrics to assess the consistency between the generated geometries and their corresponding infrared spectra. Results demonstrate that IR-GeoDiff outperforms baseline models such as EDM and GEOLDM on the QM9S dataset. Furthermore, attention visualization analysis confirms that the model's interpretation of the spectra aligns with the principles of molecular vibration derived from quantum chemical calculations.

**Strengths:**

1. The work is innovative in its application of a latent diffusion model to the challenging task of recovering molecular geometry from infrared spectra. The model architecture is novel, particularly in its use of a multi-head cross-attention mechanism for feature fusion and a Transformer-based classifier for joint feature extraction.
2. The model design is sound and the experimental evaluation is comprehensive, encompassing both spectral and structural dimensions. The inclusion of attention visualization provides valuable validation for the model's interpretability.
3. The paper is well-structured, with detailed method descriptions. The inclusion of diagrams and formulas aids in comprehension.
4. This research addresses a significant gap by enabling the direct recovery of 3D structures from infrared spectra. It holds considerable potential for applications in computational chemistry and drug design.

**Weaknesses:**

1. The experimental setup has limitations. The chosen baseline models, EDM and GEOLDM, were proposed approximately three years ago. Given the rapid pace of development in this field, more recent and potentially superior models should be included for comparison.The speed of diffusion models is too slow, and in recent years, many new models have been developed. These new models should be compared.
2. The study relies on a single dataset, QM9S, for validation. This dataset is limited to small molecules (with a maximum of nine heavy atoms), which restricts the complexity and diversity of the tested structures and may not adequately demonstrate the model's generalizability.
3. The conclusion is underdeveloped. It primarily reiterates the main contributions and findings without a critical discussion of the study's limitations or a clear outlook on future research directions.
4. The related work section lacks depth. It emphasizes the primary motivation for developing IR-GeoDiff but does not sufficiently elaborate on the core challenges or the specific innovations designed to overcome them, such as the multi-head cross-attention mechanism for fusing spectral and geometric features.
5. There are inconsistencies between the manuscript text and the mathematical notation. For instance, the variable   in Equation 6 does not match the   in line 224 of the text.

**Questions:**

1. The "Experiments" section should be strengthened by benchmarking IR-GeoDiff against more recent state-of-the-art models to compellingly demonstrate its superior performance.The speed of diffusion models is too slow, and in recent years, many new models have been developed. These new models should be compared.
2. Additional experiments on datasets featuring more complex and diverse molecular structures are recommended to thoroughly validate the model's generalization capability.
3. The "Conclusion" should be expanded to explicitly outline the study's limitations and to propose specific, actionable directions for future work.
4. The "Related Work" section should be revised to provide a clearer exposition of the core challenges in this research area and to delineate how IR-GeoDiff's design, particularly its use of cross-attention, addresses these challenges.
5. A thorough review of the entire manuscript is necessary to ensure consistency between all variables in the formulas and the main text.

---

> ### Author Response · Authors · 2025-11-21
> **Rebuttal by Authors (Part 1)**
>
> We thank the reviewer x6jw  for their detailed feedback. We are encouraged that the reviewer recognizes the novelty of our model's architecture. We also appreciate the acknowledgement of our comprehensive evaluation and the paper's potential impact on computational chemistry. Below, we address the specific concerns raised.
>
> > `w1` **The experimental setup has limitations. The chosen baseline models, EDM and GEOLDM, were proposed approximately three years ago. Given the rapid pace of development in this field, more recent and potentially superior models should be included for comparison.The speed of diffusion models is too slow, and in recent years, many new models have been developed. These new models should be compared.**
> >
> > `Q1` **The "Experiments" section should be strengthened by benchmarking IR-GeoDiff against more recent SOTA models to compellingly demonstrate its superior performance. The speed of diffusion models is too slow, and in recent years, many new models have been developed. These new models should be compared.**
>
> We chose EDM and GEOLDM as baselines because these two methods are widely regarded as the foundational architectures for 3D molecular diffusion and represent the two major design paradigms in the field: diffusion directly in coordinate space (EDM) and diffusion in a learned latent space (GEOLDM). Both have become standard baselines in subsequent 3D molecular diffusion papers, making them a natural starting point for evaluating our spectrum-conditioned recovery task.
>
> We note that EDM and GEOLDM were originally proposed for de novo generation. As discussed in the related work section, recent models in 3D molecular diffusion also consider tasks such as graph-to-3D conformer prediction and conditioning on complex modalities like protein pockets. None of these methods are designed for recovering a 3D geometry from a single IR spectrum. At submission time, we therefore prioritised adapting EDM and GEOLDM, since they are well established baselines in the field and their conditioning mechanism based on feature concatenation can be extended to IR spectral features with minimal architectural modification.
>
> We thank the reviewer’s suggestion to include more recent architectures. As suggested, we therefore further compare IR-GeoDiff to two recent equivariant diffusion models:
> * **GFMDiff**: Geometric-Facilitated Denoising Diffusion Model for 3D Molecule Generation (AAAI 2024). GFMDiff incorporates both pairwise distances and triplet-wise angles in the denoising kernel to capture higher-order local geometric interactions. These multi-body patterns are closely related to functional-group-specific vibrational modes, making GFMDiff a relevant architecture to test in our IR-conditioned setting.
> * **END**: Equivariant Neural Diffusion for Molecule Generation (NeurIPS 2024). END evaluates composition and substructure conditioned generation, which require tighter control over both global composition and local motifs and are conceptually closer to our recovery task than simple property based conditioning. In addition, the authors report that its formulation can reduce the number of function evaluations required during sampling under certain parameterizations, indicating that END has the potential to achieve more efficient sampling. This makes END a particularly relevant baseline in the context of the reviewer’s concern regarding speed.
>
> At this rebuttal stage, we have trained both models for 200 epochs, and the preliminary results are shown in the table below, where we also report the result of our method trained for 200 epochs. All models are evaluated by using 30 generated samples per test case (the computationally expensive SIS metrics were not computed here). These results already indicate that recent diffusion models do not trivially transfer to IR-spectrum-to-3D recovery, reinforcing the need for architectures explicitly designed for spectrum-conditioned structure reconstruction. The full results for both models will be reported in the final version.
>
> | Model | sim$_g$ | max sim$_g$ | Mol acc(%) |
> | -------- | -------- | -------- | -------- |
> | GFMDiff | 0.280 | 0.543 | 10.0 |
> | END | 0.127 | 0.325 | 0.5  |
> | Ours | **0.515** | **0.876** | **67.5** |
>
> Regarding the concern about speed, we acknowledge that diffusion based generative models rely on iterative sampling procedures, making them computationally slower. In this work, our goal is to introduce and study the spectrum-to-geometry recovery task and to demonstrate that high spectral and structural consistency can be achieved under a  standard diffusion-based formulation. We therefore adopt the conventional sampling procedure.
>
> Improving sampling efficiency is out of the scope of this paper, and we didn't claim speed as a contribution. It can be incorporated in future work if desired. Our focus in this initial study is on establishing the task and validating the effectiveness of the proposed spectrum conditioned latent diffusion model.

---

> ### Author Response · Authors · 2025-11-21
> **Rebuttal by Authors (Part 2)**
>
> > `w2` **The study relies on a single dataset, QM9S, for validation. This dataset is limited to small molecules (with a maximum of nine heavy atoms), which restricts the complexity and diversity of the tested structures and may not adequately demonstrate the model's generalizability.**
> >
> > `Q2` **Additional experiments on datasets featuring more complex and diverse molecular structures are recommended to thoroughly validate the model's generalization capability.**
>
> While datasets used for molecular generation tasks such as GEOM-Drugs contain larger and more diverse molecular structures, they do not provide corresponding IR spectra, which are crucial for our task. Conversely, datasets utilized in prior IR-to-structure studies usually include IR spectra but lack 3D geometries.  At the time of submission, QM9S was the only available dataset we found that offers both IR spectra and corresponding 3D molecular geometries.
>
> During the rebuttal period, we identified a recently published dataset, QMe14S [R1], which contains 186,102 molecules with IR spectra and corresponding 3D geometries. QMe14S is constructed by combining the QM9S molecules with additional 56,285 molecules selected from PubChem, resulting in a dataset that spans 14 elements and covers substantially larger and more chemically diverse structures than QM9S.
>
> **As suggested, we further perform experiments on this new dataset.** Specifically, we removed all entries overlapping with the QM9S training split to avoid any data leakage. We then applied additional quality control filtering to remove molecules with problematic geometries. The resulting subset contains 53,344 molecules. We shuffled this subset, selected 1000 molecules as the test set, and split the remaining data into training and validation sets at a 95:5 ratio.
>
> We first defined 26 common functional groups and trained a spectral classifier on the combined QM9S and QMe14S training sets for 100 epochs. We then finetuned the autoencoder and diffusion model, starting from QM9S pretrained weights, on the QMe14S training and validation set. The autoencoder was trained for 150 epochs and the diffusion model was trained for 100 epochs. For evaluation, we sampled 50 molecules for each spectrum in the 1000 molecule QMe14S test set. The results are as follows:
>
> | Data set | sim$_g$ | max sim$_g$ | mol acc (%)|
> | -------- | -------- | -------- | -------- |
> | QM9S     | 0.672 | 0.969 | 90.6 |
> | QMe14S subset | 0.569 | 0.918 | 84.3 |
>
> Although QMe14S contains molecules with larger size and greater structural diversity, the performance of IR-GeoDiff decreases only moderately. Importantly, IR-GeoDiff on QMe14S still performs markedly above the EDM and GeoLDM baselines evaluated on the simpler QM9S dataset. This demonstrates that IR-GeoDiff generalizes meaningfully beyond the QM9S domain and highlights the potential of IR-guided diffusion models to scale beyond small-molecule benchmarks.
>
> [R1] Yuan M, Zou Z, Luo Y, Jiang J, Hu W. QMe14S: A Comprehensive and Efficient Spectral Data Set for Small Organic Molecules. The Journal of Physical Chemistry Letters, 16(16):3972-9, 2025.
>
> > `w3` **The conclusion is underdeveloped. It primarily reiterates the main contributions and findings without a critical discussion of the study's limitations or a clear outlook on future research directions.**
> >
> > `Q3` **The "Conclusion" should be expanded to explicitly outline the study's limitations and to propose specific, actionable directions for future work.**
>
> We would like to clarify that a detailed discussion of the study’s limitations and future directions has already been provided in Section 5.3 (Limitations). In particular, we discussed (i) the limited control over molecular conformations, (ii) the intrinsic ambiguity of IR spectra for distinguishing certain molecular scaffolds, and (iii) the need for additional spectral modalities such as NMR to provide complementary structural information and stronger constraints on 3D recovery.
>
> Due to the page limitation, the conclusion section mainly summarizes the core contributions. In the revised version, we have expanded the conclusion section, highlighted in blue, by adding the following sentence: “*By examining cases with high graph similarity but low SIS, and vice versa, we observe that the current model still has limited control over molecular conformations. In addition, IR spectra contain intrinsic ambiguity for distinguishing certain molecular scaffolds. This motivates the use of additional spectral modalities such as NMR to provide complementary structural information and stronger constraints on 3D recovery in the future work.*”

---

> ### Author Response · Authors · 2025-11-21
> **Rebuttal by Authors (Part 3)**
>
> > `w4` **The related work section lacks depth. It emphasizes the primary motivation for developing IR-GeoDiff but does not sufficiently elaborate on the core challenges or the specific innovations designed to overcome them, such as the multi-head cross-attention mechanism for fusing spectral and geometric features.**
> >
> > `Q4` **The "Related Work" section should be revised to provide a clearer exposition of the core challenges in this research area and to delineate how IR-GeoDiff's design, particularly its use of cross-attention, addresses these challenges.**
>
> We thank the reviewer for the helpful suggestion. We have revised the “Related Work” section to more clearly describe the core challenges of spectrum-to-structure recovery and to explain how the design of IR-GeoDiff addresses these challenges.
>
> In the subsection on methods for predicting molecular structure from IR spectra, we added the following text:
> "*In contrast, IR-GeoDiff directly explore the relationship between 3D molecular geometries and IR spectra. The spectral information is injected into both atomic-level node features and bond-level edge features. In addition, inspired by prior work on functional group classification from IR spectra, we further incorporate functional group features extracted from the spectra into the edge representations. Taken together, these conditioning signals more faithfully reflect the physical origin of IR absorption in the vibrations of specific bond and local patterns.*"
>
> In the subsection on diffusion models for 3D molecular generation, we added the following text:
> *"Despite this progress, there is currently no diffusion model that is conditioned on full IR spectra and learns a distribution over three-dimensional molecular geometries. Likewise, no evaluation protocol has been established for this new spectrum-to-geometry recovery task, which is fundamentally different from previous generation tasks. The objective is not to encourage diversity in the generated molecules, but to recover geometries that are consistent with a given spectrum and to reduce the candidate space as much as possible. To address these challenges, we design IR-GeoDiff as an SE(3)-equivariant latent diffusion model conditioned on spectral features. A Transformer-based spectral encoder extracts global spectral representations along with chemically meaningful functional group features, which serve as conditioning signals for the denoising network. These conditioning signals are integrated into the latent diffusion process via multi-head cross-attention. This design is tailored to the physics of IR spectroscopy, and it provides a more interpretable way to fuse spectral and geometric features. In addition, we propose a comprehensive set of evaluation metrics that assess whether the recovered geometries are consistent with the input IR spectra from both structural and spectral perspectives."*
>
> The modification above are highlighted in blue in the revised manuscript.
>
>
> > `w5` **There are inconsistencies between the manuscript text and the mathematical notation. For instance, the variable in Equation 6 does not match the in line 224 of the text.**
>
> We appreciate the reviewer for pointing out this typo. We have corrected it by changing $\tilde{\mathrm{y}}_i$ to $\tilde{\mathrm{y}}_k$, so that the notation is now consistent with Equation (6).
>
>
> > `Q5` **A thorough review of the entire manuscript is necessary to ensure consistency between all variables in the formulas and the main text.**
>
> We thank the reviewer for this reminder. We have carefully rechecked the entire manuscript to ensure that the variables used in the formulas and in the main text are consistent. All necessary corrections have been made, and the corresponding changes are highlighted in blue in the revised version.

---

### Official Review · Reviewer_xKpc · 2025-11-02

**Soundness:** 2
**Presentation:** 2
**Contribution:** 2
**Rating:** 2
**Confidence:** 4

**Summary:**

This paper proposes IR-GeoDiff, a latent diffusion model designed to recover 3D molecular geometries from infrared (IR) vibrational spectra. The model integrates spectral features into both node and edge representations within an equivariant latent diffusion framework. Experiments on the QM9S dataset demonstrate that IR-GeoDiff can reconstruct 3D structures with given IR spectra, achieving promising results on authors' designed evaluation metrics.

**Strengths:**

1. The idea of reconstructing 3D molecular geometries from vibrational spectra is scientifically interesting and potentially impactful for computational chemistry and molecular spectroscopy.
2. The paper is overall well written and logically organized. It's easy to follow and recognize the paper's contributions.

**Weaknesses:**

1. The motivation for adopting a latent diffusion model is not sufficiently convincing. Latent diffusion models rely heavily on strong VAE encoders/decoders to build meaningful latent representations, but such powerful autoencoders are not yet well established for molecular 3D geometry. In contrast, existing non-latent 3D molecular diffusion models already achieve strong and stable results directly in coordinate space. The paper should clearly justify why the latent-space formulation is preferable in this domain.
2. The experiments only compare with EDM and GEOLDM. More recent and stronger baselines, including non-latent 3D molecular diffusion models, should be included for a fair evaluation.
3. The claimed contribution focuses on 3D molecular generation, but the primary structural metric is the Tanimoto similarity between Morgan fingerprints. This metric does not capture 3D conformational or geometric correctness. Evaluations for quality of generated 3D structures should be reported.
4. The method assumes that the atom types and atom count are known a priori. Although the authors justify this as reflecting certain experimental workflows, this assumption greatly limits practical usability. In real scenarios, these properties may be unknown or partially uncertain, making the proposed method inapplicable beyond constrained settings.

**Questions:**

1. Is the autoencoder module trained by the authors, or reused from prior molecular latent diffusion work (e.g., GEOLDM)? Please clarify its training procedure, data, and reconstruction performance.
2. When computing *molecular accuracy*, how is a correct molecule defined? Is it based on exact molecular graph structure, fingerprint, or SMILES?
3. Since the paper reports top-$n$ accuracy, what value of $n$ is used, and how sensitive are results to this choice?

---

> ### Author Response · Authors · 2025-11-21
> **Rebuttal by Authors (Part 1)**
>
> We thank Reviewer xKpc for their time and effort in reviewing our work. We appreciate that the reviewer recognizes the scientific interest and potential impact of our proposed task, as well as the clarity and logical organization of the manuscript. Below we respond to the specific concerns raised.
>
> > `w1`. **The motivation for adopting a latent diffusion model is not sufficiently convincing. Latent diffusion models rely heavily on strong VAE encoders/decoders to build meaningful latent representations, but such powerful autoencoders are not yet well established for molecular 3D geometry. In contrast, existing non-latent 3D molecular diffusion models already achieve strong and stable results directly in coordinate space. The paper should clearly justify why the latent-space formulation is preferable in this domain.**
>
> We respectfully disagree with the statement that “powerful autoencoders are not yet well established for 3D molecular geometry.” Recent work, most notably GEOLDM (Xu et al., 2023), directly addresses this challenge and provides a principled, SE(3)-equivariant autoencoding framework specifically designed for molecular geometries. GEOLDM constructs a latent space consisting of both invariant scalars and equivariant tensors, ensuring strict preservation of roto-translational symmetry. Our method builds upon this established and carefully validated framework rather than relying on an ad-hoc encoder–decoder design.
>
> In addition, our recovery task **fundamentally differs from molecular generation**. Recovering a molecular geometry from an IR spectrum requires strong controllability: given an IR spectrum, the goal is to reduce the candidate space as much as possible, not to promote diversity, as in existing generative models. The latent-diffusion formulation offers two advantages in this setting:
> * Latent representation provides a smoother diffusion space: The autoencoder maps the raw, multi-modal atomic space (discrete atom types and continuous coordinates) into a compact latent space that is better behaved for Gaussian diffusion. Xu et al. (2023) show that this latent space preserves roto-translational symmetry and leads to more stable training compared with directly diffusing on raw coordinates.
> * Latent modeling improves controllability: The use of latent variables also allows for better control over the generation process (Rombach et al., 2022). Additionally,  Xu et al. (2023) demonstrate GEOLDM’s higher capacity for controllable generation just due to the latent modeling.
>
> Finally, our experimental results further supports the choice of the latent formulation. Under the same conditioning setup, GEOLDM consistently outperforms EDM on all metrics relevant to the recovery task (Table 1), indicating that latent-space modeling is more suitable for spectrum-guided 3D geometry recovery.

---

> > ### Author Response · Authors · 2025-11-21
> > **Rebuttal by Authors (Part 2)**
> >
> > > `w2` **The experiments only compare with EDM and GEOLDM. More recent and stronger baselines, including non-latent 3D molecular diffusion models, should be included for a fair evaluation.**
> >
> > We chose EDM and GEOLDM as baselines because these two methods are widely regarded as the foundational architectures for 3D molecular diffusion and represent the two major design paradigms in the field: diffusion directly in coordinate space (EDM) and diffusion in a learned latent space (GEOLDM). Both have become standard baselines in subsequent 3D molecular diffusion papers, making them a natural starting point for evaluating our spectrum-conditioned recovery task.
> >
> > We note that EDM and GEOLDM were originally proposed for unconditional or property-conditioned de novo generation. As discussed in the related work section, recent models in 3D molecular diffusion also consider tasks such as graph-to-3D conformer prediction and conditioning on complex geometric modalities including protein pockets or reference 3D structures. None of these methods are designed for recovering a 3D molecular geometry from a single IR spectrum. At submission time, we therefore prioritised adapting EDM and GEOLDM, since they are well established baselines in the field and their conditioning mechanism based on feature concatenation can be extended to IR spectral features with minimal architectural modification.
> >
> > We thank the reviewer’s suggestion to include more recent architectures. As suggested, we therefore further compare IR-GeoDiff to two recent equivariant diffusion models:
> > * **GFMDiff**: Geometric-Facilitated Denoising Diffusion Model for 3D Molecule Generation (AAAI 2024). GFMDiff incorporates both pairwise distances and triplet-wise angles in the denoising kernel to capture higher-order local geometric interactions. These multi-body patterns are closely related to functional-group-specific vibrational modes, making GFMDiff a relevant architecture to test in our IR-conditioned setting.
> > * **END**: Equivariant Neural Diffusion for Molecule Generation (NeurIPS 2024). END is an Euclidean-equivariant diffusion model with a learnable forward process. END evaluates composition-conditioned and substructure-conditioned generation, which require tighter control over both global composition and local motifs and are thus conceptually closer to our spectrum-guided recovery task than simple property-based conditioning.
> >
> > At this rebuttal stage, we have trained both models for 200 epochs, and the preliminary results are shown in the table below, where we also report the result of our method trained for 200 epochs. All models are evaluated by using 30 samples per test case (the computationally expensive SIS metrics are not computed here). These results already indicate that recent diffusion models do not trivially transfer to IR-spectrum-to-3D recovery, reinforcing the need for architectures explicitly designed for spectrum-conditioned structure reconstruction. The full results for both models will be reported in the final version.
> >
> > | Model (trained 200 epochs) | sim$_g$ | max sim$_g$ | Mol acc(%) |
> > | -------- | -------- | -------- | -------- |
> > | GFMDiff | 0.280 | 0.543 | 10.0 |
> > | END | 0.127 | 0.325 | 0.5  |
> > | Ours | **0.515** | **0.876** | **67.5** |

---

> ### Author Response · Authors · 2025-11-21
> **Rebuttal by Authors (Part 3)**
>
> > `w3` **The claimed contribution focuses on 3D molecular generation, but the primary structural metric is the Tanimoto similarity between Morgan fingerprints. This metric does not capture 3D conformational or geometric correctness. Evaluations for quality of generated 3D structures should be reported.**
>
> We would like to clarify that the quality of generated 3D structures **has been explicitly evaluated in our submitted paper**. Appendix E.4 reports validity, stability and connectivity, which are standard metrics in the 3D molecular generation literature. These metrics assess whether a generated geometry is physically meaningful.
>
> Beyond these metrics, spectral information similarity (SIS) serves as a physically grounded measure of geometric correctness in our task. IR spectra are produced through quantum mechanical vibrational analysis that takes the full 3D geometry as input. Any geometric or conformational discrepancy will manifest in the resulting spectrum. Consequently, an inaccurate 3D geometry will lead to a mismatched spectrum and a lower SIS score. In Section 5.3 and Figure 4C, we analyzed cases where subtle conformational differences, such as hydrogen bond formation, reduce SIS even when the molecular graph remains unchanged. These examples show that SIS is highly sensitive to geometric fidelity. Besides, SIS directly evaluates this alignment between geometry and spectrum and is therefore the task-appropriate measure of 3D correctness.
>
> Finally, the use of Tanimoto similarity provides a complementary and chemically meaningful view of structural correctness. This metric reflects whether the overall molecular graph, including atomic connectivity and functional-group patterns, is recovered accurately.
>
> We have updated the manuscript to make this point explicit. In the revised version, we added the following sentence, highlighted in blue: “*SIS also serves as a physically grounded measure of geometric correctness, since IR spectra are computed directly from the full three-dimensional geometry.*”
>
>
> > `w4` **The method assumes that the atom types and atom count are known a priori. Although the authors justify this as reflecting certain experimental workflows, this assumption greatly limits practical usability. In real scenarios, these properties may be unknown or partially uncertain, making the proposed method inapplicable beyond constrained settings.**
>
> We respectfully disagree with the claim that assuming access to the atomic composition “greatly limits practical usability.”
>
> In real spectroscopic workflows, IR spectroscopy is primarily applied for qualitative analysis, especially for identifying the presence of specific functional groups (Coates et al., 2000, Clayden et al., 2012), and it provides limited information about the full atomic composition, making it extremely difficult to determine the complete molecular structure based on IR data alone. Therefore, in practice, structure elucidation is rarely attempted using IR spectra in isolation. Specifically, as outlined in [R1] (Chapter 8, Determining the Structure of Organic Molecules from Spectra), the molecular formula is typically established beforehand using other techniques such as elemental analysis or complementary spectroscopic data, and is then used as a known condition when interpreting IR spectra (Line 158-161). Our assumption therefore reflects a realistic, standard, and experimentally grounded setting rather than an artificial constraint.
>
> Moreover, conditioning on the atomic composition is fully consistent with existing work on spectrum-to-structure recovery. Because a single spectrum rarely contains enough information to uniquely determine molecular identity, it is common practice to use the molecular formula as an input condition. This is the case for IR spectrum conditioned structure prediction (Alberts et al., 2024a; Wu et al., 2025; Alberts et al., 2025) (Line 209-210), for mass spectrometry conditioned molecular generation (Bohde et al., 2025) (Line129-130), and also for NMR spectrum inverse problems [R2]. In all of these settings, the atomic composition is treated as part of the available prior information, and the model focuses on resolving the remaining structural ambiguity given that composition.
>
> In summary, **our formulation aligns with how spectra are actually used in laboratory workflows and with how prior spectrum conditioned generative models are designed in the literature.** While our method is not intended to replace all upstream analytical steps required to determine the molecular formula, the assumption of known atomic composition does not severely limit practical usability. Instead, it represents a necessary and standard component of any realistic spectrum-to-structure pipeline.
>
> [R1] Field, L, et al. Organic structures from spectra. John Wiley & Sons, 2013.
>
> [R2] Jonas, Eric. "Deep imitation learning for molecular inverse problems." NeurIPS 2019.

---

> > ### Author Response · Authors · 2025-11-21
> > **Rebuttal by Authors (Part 4)**
> >
> > > `Q1` **Is the autoencoder module trained by the authors, or reused from prior molecular latent diffusion work (e.g., GEOLDM)? Please clarify its training procedure, data, and reconstruction performance.**
> >
> > The autoencoder is fully trained by us from scratch on the QM9S dataset, using the same training and validation splits as the spectral classifier and the diffusion model. We do not reuse any pretrained autoencoder from prior latent diffusion work such as GEOLDM. Our design only follows the architectural specification of GEOLDM, namely an EGNN-based encoder and decoder.
> >
> > The training procedure was described in Section 4.4 and Appendix B.3. We first pretrain the spectral classifier using the functional group classification loss $L_{cls}$. In the next stage, we jointly optimise the autoencoder and the classifier with the objective $L_{AE}+ L_{cls}$. During diffusion model training, the classifier is frozen so that it provides a stable conditioning signal, while the autoencoder continues to be optimised together with the diffusion model using $L_{AE} + L_{LDM}$.
> >
> > To address the reviewer’s question regarding reconstruction performance, we further evaluate the autoencoder on the held-out QM9S test set. The mean squared error (MSE) between the input coordinates and the reconstructed geometry is:
> >
> > |  | After AE training (Å$^2$) | After DM training (Å$^2$) |
> > | -------- | -------- | -------- |
> > | MSE (test set)  | $2.98\times10^{-3}$     |$4.79\times10^{-6}$|
> >
> > The autoencoder achieves low reconstruction error after pretraining, and the error further decreases after the full diffusion model is trained jointly with the autoencoder.
> >
> >
> > > `Q2` **When computing molecular accuracy, how is a correct molecule defined? Is it based on exact molecular graph structure, fingerprint, or SMILES?**
> >
> > Following previous IR-to-structure work (Alberts et al., 2024a; Wu et al., 2025; Alberts et al., 2025), we define a molecule as correct if its canonical SMILES string exactly matches the canonical SMILES of the reference molecule. Canonical SMILES provides a unique representation of the molecular graph, which allows us to determine chemical identity in a consistent and unambiguous manner across all generated samples.
> >
> > > `Q3` **Since the paper reports top-n accuracy, what value of n is used, and how sensitive are results to this choice?**
> >
> > We would like to clarify that our method does not use top-n accuracy. Prior IR-to-SMILES work (for example Alberts et al., 2024a and Wu et al., 2025) reports top-n accuracy because the Transformer-based models can produce an ordered list of n candidate SMILES through beam search. In contrast, diffusion models generate 3D structures through stochastic sampling and do not provide a ranked list of outputs.
> >
> > In our main experiments, we used 50 samples per spectrum. The sensitivity of *molecular accuracy* to this choice was reported in Appendix E.3 and referenced in Line 346. As shown in Figure A4, the *molecular accuracy* increases rapidly and becomes stable once *the number of samples* exceeds approximately 30. This indicates that using 50 samples provides a reliable and representative estimate of structure recovery performance.

---

### Author Response · Authors · 2025-12-02
**General Response**

We thank all reviewers and ACs for their time and effort. We are grateful for the reviewers' recognition of the novelty of our proposed task: recovering 3D molecular geometries directly from a 1D infrared spectrum.

To address this challenging problem, we introduced IR-GeoDiff, an SE(3)-equivariant latent diffusion model, together with a comprehensive evaluation protocol combining spectral and structural similarity. We appreciate the reviewers' thoughtful feedback to give us an opportunity to clarify our work.
* IR-GeoDiff incorporates task-specific innovations motivated by the physics of infrared spectroscopy, rather than being a simple combination of existing components. The cross-attention visualizations show that the model’s behavior aligns with how chemists interpret IR spectra and with quantum-vibrational principles, offering new insights into model interpretability (reviewers YN2k-w3 and sQdo-w1).
* The deviation in recovered 3D geometries is especially measured by spectral similarity (SIS) as discussed in Section 5.3 (reviewers xKpc-w3 and YN2k-Q4).
* The assumption of a known molecular formula is standard in practical analytical workflows and consistent with prior single-spectrum studies (reviewers xKpc-w4 and sQdo-w2).

In this rebuttal, we further conduct additional experiments to address reviewers' concerns.
* We added GFMDiff and END as two recent, stronger baselines for a more comprehensive comparison (reviewers xKpc-w2, x6jw-w1&Q1, and YN2k-w4).
* We performed EDM and GeoLDM variant experiments to directly address the concern about fixing atom counts while leaving atom types unconstrained (reviewer YN2k-w5&Q2&follow-up).
* We evaluated IR-GeoDiff on the QMe14S dataset, demonstrating meaningful generalization to larger and more chemically diverse molecules (reviewers x6jw-w2&Q2, and YN2k-w2).
* We reported the reconstruction performance of the autoencoder, showing that keeping the autoencoder learnable during diffusion training further improves the latent representation (reviewers xKpc-Q1 and YN2k-Q1).

All revisions in the manuscript are highlighted in blue. Full-scale results for the additional experiments will be reported in the final version for completeness.

We again thank the reviewers for their thoughtful comments, which have significantly improved the clarity and quality of the work.

---

### Meta-Review · Area_Chair_2fwz · 2025-12-29

**Summary:**

The paper proposes a new method for determining molecular structures from infrared (IR) spectra leveraging diffusion models. As such, the paper introduces both a new task, including evaluation metrics for the task, along with IR-GeoDiff, an SE(3)-equivariant latent diffusion model to solve the proposed task of IR to structure mapping.

The reviewers generally appreciated the importance of the IR-based molecular structure recovery and noted helpful explanations in the paper relating to the method and some of the background materials. Being able to identify and recover molecular structures from IR spectra would be an impactful machine learning application to chemistry and provide room for innovation in both machine learning techniques and benchmarking.

While the presentation of the task is relevant and timely, the reviewers expressed concerns about the benchmarking and justification of the baselines and information available to the model. The questions on the benchmarking, in addition to various clarifications sought by the reviewers, led to extensive discussion with the authors. The results of this discussion were many clarifying points along the choices made for the benchmarking and evaluation, including how data was selected and split and what procedures are needed to obtain proper labels. On top of that, the authors provided additional experimental results benchmarking additional models on the existing data in the paper (QM9-S) and new results on a new dataset (QMe14S dataset).

While the authors made significant efforts in the rebuttal and I believe the clarifications and edits discussed would improve the paper significantly, I think that the paper would benefit from a comprehensive re-writing that would emphasize the benchmarking choices and procedures to properly set up the task and data. With this, the authors could provide additional contributions to the community in terms of enabling other researchers to develop methods for IR-based molecular recovery. The technical work itself is relevant and promising, and a future revision of the paper with reviewer feedback included would be significantly clearer and stronger.

**Reviewer Concerns:**

Addressed Concerns:
* Reviewer xKpc's concerns on justifying the use of latent diffusion models.
* Reviewer x6jw's concerns on providing additional details on related work.
* Reviewer YN2k's concerns on clarifying the problem setting, method details and data used.
* Reviewer sQdo's concerns on assumptions of the method inputs and novelty of the method.


Outstanding Concerns:
* Reviewer xKpc's concern on more robust benchmarking. The inclusion of additional methods and more datasets only partially address this - the authors should better motivate their benchmark and evaluation choices.
* Reviewer x6jw's concerns on benchmarking appear partially addressed similar to the benchmarking concerns raised by all reviewers.
* Reviewer YN2k' concerns on benchmarking and baseline choices as well as certain method details.
* Reviewer sQdo's concerns on benchmarking and evaluation metrics are partially addressed.

**Reviewer Scores:**

* Reviewer xKpc remains at 2 or maybe raises to 4.
* Reviewer x6jw remains at 4.
* Reviewer YN2k probably raises to 4.
* Reviewer sQdo remains at 4.

Overall, even with increased scores none of the reviewers would support the paper.

---

### Decision · Program_Chairs · 2026-01-26

Reject